# VIPAMIN: Visual Prompt Initialization via Embedding Selection and Subspace Expansion

**Jaekyun Park**
School of Electrical Engineering
KAIST
jaekyun.park@kaist.ac.kr

**Hye Won Chung**$^*$
School of Electrical Engineering
KAIST
hwchung@kaist.ac.kr

## Abstract

In the era of large-scale foundation models, fully fine-tuning pretrained networks for each downstream task is often prohibitively resource-intensive. Prompt tuning offers a lightweight alternative by introducing tunable prompts while keeping the backbone frozen. However, existing visual prompt tuning methods often fail to specialize the prompts or enrich the representation space–especially when applied to self-supervised backbones. We show that these limitations become especially pronounced in challenging tasks and data-scarce settings, where effective adaptation is most critical. In this work, we introduce VIPAMIN, a visual prompt initialization strategy that enhances adaptation of self-supervised models by (1) aligning prompts with semantically informative regions in the embedding space, and (2) injecting novel representational directions beyond the pretrained subspace. Despite its simplicity–requiring only a single forward pass and lightweight operations–VIPAMIN consistently improves performance across diverse tasks and dataset sizes, setting a new state of the art in visual prompt tuning. Our code is available at https://github.com/iamjaekyun/vipamin.

## 1 Introduction

Large-scale Vision Transformers (ViTs) have become central to modern computer vision. Traditionally, adapting a pretrained model to a new downstream task involves full fine-tuning, updating all model parameters and a task-specific head. However, as foundation models continue to scale, full fine-tuning incurs substantial computational and memory overhead.

Prompt tuning offers a more efficient alternative by introducing a small number of trainable tokens (prompts) while keeping the backbone frozen. Initially successful in NLP, prompt tuning has shown strong performance in vision through Visual Prompt Tuning (VPT) [27], which rivals full fine-tuning on many benchmarks using only a few dozen tokens [27, 36, 37, 58]. VPT has also proven effective in more challenging settings such as test-time adaptation [47] and continual learning [26, 38, 67].

While recent theoretical work shows that prompt tokens can selectively modulate attention to emphasize relevant inputs [48], and may serve as retrieval mechanisms for pretrained knowledge [51, 65], these findings do not fully explain VPT's limitations in practice–especially with self-supervised backbones. Empirically, we identify two key failure modes in such settings: (1) prompts exhibit near-uniform attention across tokens, lacking specialization, and (2) prompt outputs collapse into the pretrained self-attention subspace, failing to introduce novel task-relevant features. These limitations severely hinder VPT's adaptability, especially under distribution shifts and in low-data regimes where representational diversity and efficient adaptation are essential.

---

$^*$Corresponding author.

39th Conference on Neural Information Processing Systems (NeurIPS 2025).

Motivated by these findings, we introduce VIPAMIN: **V**isual **I**nitialization of **P**rompts via **A**ttention-guided **M**atching and **I**njection of **N**ovelty. VIPAMIN is a lightweight initialization scheme requiring only a single forward pass through the model and two inexpensive matrix operations. It comprises two components designed to address the key failure modes observed in vanilla VPT. The matching module aligns each prompt with semantically coherent input regions, mitigating the issue of uniform attention. The orthogonalizing module projects the prompt away from the row space of the frozen self-attention output, introducing novel representational directions and alleviating subspace collapse. VIPAMIN adds no additional learnable parameters, incurs minimal computational overhead, and integrates easily into existing VPT pipelines with a one-time initialization step.

We rigorously benchmark VIPAMIN on 19 vision tasks–covering natural, specialized, and structured recognition–using two leading self-supervised backbones (MoCo-v3 [7] and MAE [21]). Additionally, we assess its effectiveness under data-scarce conditions on five few-shot benchmarks. Concretely, for MoCo-v3 pretrained model, VIPAMIN achieves a 3.7% gain over full fine-tuning on tasks that deviate significantly from the pretraining distribution (i.e., Structured), and a 4.6% gain on tasks more closely aligned with the pretraining domain (i.e., Natural)–establishing the new state-of-the-art performance for visual prompt tuning without introducing additional parameters, computational latency, or memory overhead. In few-shot settings, VIPAMIN consistently outperforms the baseline across all datasets. Our contributions are summarized as follows:

- We identify two underexplored limitations of self-supervised VPT–uniform attention and subspace collapse–arising from the internal dynamics of the transformer architecture.

- We introduce VIPAMIN, a lightweight, non-intrusive initialization technique that addresses these issues via attention-guided matching and orthogonal subspace injection.

- We demonstrate that VIPAMIN is readily deployable, computationally efficient, and consistently enhances performance across a wide range of datasets,

**Related Work**    VPT has shown suboptimal performance when applied to self-supervised models. GatedPT improves visual prompting by facilitating inter-block interactions of ViT, demonstrating effectiveness in such self-supervised settings  [70]. The work on SPT, most closely related to ours, shows that prompt representations tend to collapse onto the embedding tokens during training and actively leverages this property by initializing prompts with downstream token prototypes to improve VPT's convergence [66]. iVPT enhances prompt–embedding interplay via an attentive reinforcement module that steers prompts toward salient tokens [76], while VFPT reduces the domain gap by modulating a subset of prompts in the frequency domain using a 2D Fast Fourier Transform [72]. Most recently, DA-VPT selects prompt membership based on the CLS token of same-class images, optimizing the assignments via metric learning [53].

Unlike prior methods that add learnable gates (GatedPT), reinforcement blocks (iVPT), metric learning (DA-VPT), or Fourier transforms (VFPT), VIPAMIN modifies only the initial prompt weights–incurring zero overhead. Whereas SPT also incurs no additional cost in training, its accuracy hinges on a costly offline clustering step that is hard to amortize, while VIPAMIN exceeds the performance of SPT with only a couple of lightweight matrix operations.

## 2    Preliminaries: Roles of Prompts in Visual Prompt Tuning

We use the following notations throughout the paper. A bold lowercase character (e.g., $\mathbf{x}$) denotes a vector, while a bold uppercase character (e.g., $\mathbf{P}$) denotes a matrix. The $i$-th row and the $(i, j)$-th entry of the matrix $\mathbf{P}$ are represented by $(\mathbf{P})_i$ and $(\mathbf{P})_{ij}$, respectively. Let $(\cdot, \cdot)$ denote column-wise concatenation, and $[\cdot; \cdot]$ denote row-wise concatenation. Let $[n] := \{i | i \in \mathbb{N}, 1 \leq i \leq n\}$.

**ViT and VPT**    ViT [13] processes an input image by dividing it into a fixed number of non-overlapping patches, each corresponding to a token in the transformer. For an image $\mathbf{x} \in \mathbb{R}^{3 \times h \times w}$ and a patch size $(h', w')$, the number of resulting tokens is $N_e := \frac{h}{h'} \times \frac{w}{w'} + 1$, including the additional 'class token' (CLS) used in image classification. These tokens are passed through a PatchEmbed module and added by learnable positional embeddings. We denote the resulting embedding matrix for the $N_e$ input tokens of a sample $\mathbf{x}$ as $\mathbf{E}_0 = (\mathbf{x}_1, \ldots, \mathbf{x}_{N_e})^\top \in \mathbb{R}^{N_e \times d}$.

VPT [27] introduces a set of learnable prompts $\mathbf{P}_0 = (\mathbf{p}_1, \ldots, \mathbf{p}_{N_p})^\top \in \mathbb{R}^{N_p \times d}$ prepended to the input image tokens $\mathbf{E}_0$, enabling lightweight task adaptation while keeping the original ViT frozen. The combined input to the transformer is denoted by $\mathbf{Z}_0 = [\mathbf{P}_0; \mathbf{E}_0] \in \mathbb{R}^{(N_p + N_e) \times d}$. VPT has two common variants: VPT-Shallow, in which the prompt $\mathbf{P}_0$ is prepended before the first transformer block, and VPT-Deep, which injects distinct, learnable prompts before each transformer block. In this section, we focus on VPT-Shallow and analyze the propagation of the initial prompt $\mathbf{P}_0$ through the ViT architecture. Assuming the ViT consists of $L$ blocks, let $B_i$ denote the operation of the $i$-th block. The evolution of the prompt and input embeddings through each block is given by:

$$[\mathbf{P}_i; \mathbf{X}_i] = B_i([\mathbf{P}_{i-1}; \mathbf{X}_{i-1}]) \in \mathbb{R}^{(N_p + N_e) \times d} \text{ for } i \in [L], \tag{1}$$

where $\mathbf{X}_0 = \mathbf{E}_0$. Each block consists of a Multi-head Self-Attention (MSA) module followed by a Feed-Forward Network (FFN), with Layer Normalization (LN) and residual connections. For simplicity, we focus on a single-head Self-Attention (SA) applied to prompt-prepended input $\mathbf{Z}_0$:

$$\text{SA}(\mathbf{Z}_0) = \text{softmax}\left(\frac{\mathbf{Z}_0 \mathbf{W}_Q (\mathbf{Z}_0 \mathbf{W}_K)^\top}{\sqrt{d}}\right) \mathbf{Z}_0 \mathbf{W}_V \in \mathbb{R}^{(N_p + N_e) \times d}, \tag{2}$$

where $\mathbf{W}_Q, \mathbf{W}_K, \mathbf{W}_V \in \mathbb{R}^{d \times d}$ are the query, key, and value projection matrices, respectively. The softmax is applied row-wise to normalize the attention weights.

To better understand the role of $\mathbf{P}_0$, we decompose the matrices involved. For $\mathbf{Z}_0 = [\mathbf{P}_0; \mathbf{X}_0]$, let

$$\mathbf{Q} = \mathbf{Z}_0 \mathbf{W}_Q = [\mathbf{Q}_{\mathbf{P}_0}; \mathbf{Q}_{\mathbf{X}_0}]; \quad \mathbf{K} = \mathbf{Z}_0 \mathbf{W}_K = [\mathbf{K}_{\mathbf{P}_0}; \mathbf{K}_{\mathbf{X}_0}]; \quad \mathbf{V} = \mathbf{Z}_0 \mathbf{W}_V = [\mathbf{V}_{\mathbf{P}_0}; \mathbf{V}_{\mathbf{X}_0}].$$

The attention matrix $\mathbf{S}$ and the resulting SA output (2) can then be written as:

$$\mathbf{S} = \text{softmax}\left(\frac{\mathbf{Q}\mathbf{K}^\top}{\sqrt{d}}\right) = \text{softmax}\left(\frac{1}{\sqrt{d}}\begin{bmatrix} \mathbf{Q}_{\mathbf{P}_0}\mathbf{K}_{\mathbf{P}_0}^\top & \mathbf{Q}_{\mathbf{P}_0}\mathbf{K}_{\mathbf{X}_0}^\top \\ \mathbf{Q}_{\mathbf{X}_0}\mathbf{K}_{\mathbf{P}_0}^\top & \mathbf{Q}_{\mathbf{X}_0}\mathbf{K}_{\mathbf{X}_0}^\top \end{bmatrix}\right) := \begin{bmatrix} \mathbf{S}_{\mathbf{P}_0\mathbf{P}_0} & \mathbf{S}_{\mathbf{P}_0\mathbf{X}_0} \\ \mathbf{S}_{\mathbf{X}_0\mathbf{P}_0} & \mathbf{S}_{\mathbf{X}_0\mathbf{X}_0} \end{bmatrix}; \tag{3}$$

$$\text{SA}(\mathbf{Z}_0) = \begin{bmatrix} \mathbf{S}_{\mathbf{P}_0\mathbf{P}_0} & \mathbf{S}_{\mathbf{P}_0\mathbf{X}_0} \\ \mathbf{S}_{\mathbf{X}_0\mathbf{P}_0} & \mathbf{S}_{\mathbf{X}_0\mathbf{X}_0} \end{bmatrix}\begin{bmatrix} \mathbf{P}_0\mathbf{W}_V \\ \mathbf{X}_0\mathbf{W}_V \end{bmatrix} = \begin{bmatrix} \mathbf{S}_{\mathbf{P}_0\mathbf{P}_0}(\mathbf{P}_0\mathbf{W}_V) + \mathbf{S}_{\mathbf{P}_0\mathbf{X}_0}(\mathbf{X}_0\mathbf{W}_V) \\ \mathbf{S}_{\mathbf{X}_0\mathbf{P}_0}(\mathbf{P}_0\mathbf{W}_V) + \mathbf{S}_{\mathbf{X}_0\mathbf{X}_0}(\mathbf{X}_0\mathbf{W}_V) \end{bmatrix} \in \mathbb{R}^{(N_p + N_e) \times d}.$$

Note that in the absence of prompts $\mathbf{P}_0$, the self-attention output for the original input tokens $\mathbf{X}_0$ reduces to $\text{SA}(\mathbf{X}_0) = \mathbf{S}_{X_0}(\mathbf{X}_0\mathbf{W}_V) \in \mathbb{R}^{N_e \times d}$, where $\mathbf{S}_{\mathbf{X}_0}$ denotes the attention weights computed solely among the input tokens. Comparing this to the prompt-augmented case reveals two distinct roles played by prompts in modifying the output of the self-attention module: (1) The prompts $\mathbf{P}_0$ attend to the input tokens $\mathbf{X}_0$, filtering, selecting, and aggregating relevant information through $\mathbf{S}_{\mathbf{P}_0\mathbf{X}_0}(\mathbf{X}_0\mathbf{W}_V)$, which is then passed to the next block via the prompt outputs. (2) In addition, the prompts inject new information into the input token representations by contributing a prompt-induced bias term $\mathbf{S}_{\mathbf{X}_0\mathbf{P}_0}(\mathbf{P}_0\mathbf{W}_V)$, thereby modifying the self-attention output for the original tokens.

**Role 1: Selecting Tokens for Prompt Propagation** The first $N_p$ rows of $\text{SA}(\mathbf{Z}_0)$ correspond to the updated prompt representations after a ViT block. As described in (1), these outputs are propagated through the transformer in VPT-Shallow. The effectiveness of VPT hinges on the prompts' ability to selectively aggregate and transmit informative content from the input tokens. Each row of $\mathbf{S}_{\mathbf{P}_0\mathbf{X}_0}(\mathbf{X}_0\mathbf{W}_V) \in \mathbb{R}^{N_p \times d}$ forms a weighted combination of input embeddings, where the weights are given by the attention scores in $\mathbf{S}_{\mathbf{P}_0\mathbf{X}_0}$. Since only a subset of input tokens are typically task-relevant, selective attention is essential for filtering noise and propagating meaningful information.

**Role 2: Injecting Semantic Bias into Attention Outputs** Prompts also modulate the output of input tokens by introducing a prompt-induced bias. The updated self-attention output for input tokens (the last $N_e$ rows of $\text{SA}(\mathbf{Z}_0)$) is given by:

$$\mathbf{S}_{\mathbf{X}_0\mathbf{P}_0}(\mathbf{P}_0\mathbf{W}_V) + \mathbf{S}_{\mathbf{X}_0\mathbf{X}_0}(\mathbf{X}_0\mathbf{W}_V) = \mathbf{S}_{\mathbf{X}_0\mathbf{P}_0}(\mathbf{P}_0\mathbf{W}_V) + \text{diag}(\mathbb{1}_{N_e} - \mathbf{S}_{\mathbf{X}_0\mathbf{P}_0}\mathbb{1}_{N_p})\text{SA}(\mathbf{X}_0), \tag{4}$$

where $\text{SA}(\mathbf{X}_0)$ is the self-attention output without prompts. This shows that prompts inject a linear bias into the attention output, potentially shifting representations toward task-relevant semantics [25, 51]. In the next section, we empirically evaluate whether VPT effectively fulfills these two roles–semantic token selection and semantic bias injection–especially in challenging downstream tasks and settings.

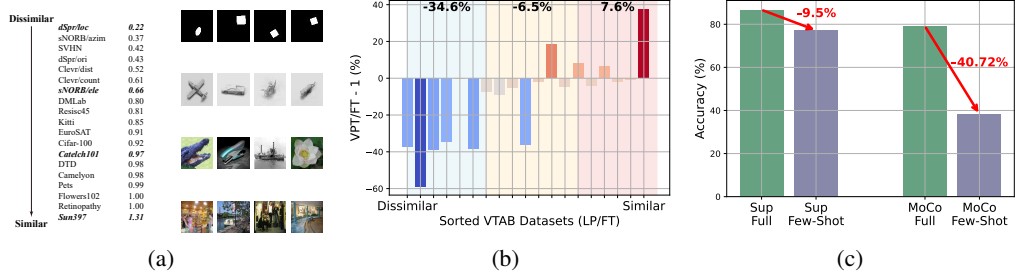

Figure 1: (a) Nineteen VTAB-1k tasks ordered by a proxy for task similarity to the pretraining dataset, measured by the relative accuracy ratio of linear probing compared to full fine-tuning (LP ratio). Representative images from four datasets—dSprites/loc, smallNORB/azi, Caltech101, and Sun397—are shown to the right. (b) Relative accuracy ratio of VPT compared to full fine-tuning (VPT ratio) across 19 VTAB datasets, sorted by the LP ratio. (c) Accuracy comparison of VPT between training with the full dataset and few-shot learning (8 images per class), evaluated on the CUB-200-2011 dataset using supervised and self-supervised (MoCo-v3) pretrained weights.

## 3    Motivation

To evaluate whether prompt tuning effectively leverages its two core mechanisms–semantic token selection and semantic bias injection–we focus on two settings where these capabilities are especially critical: (i) distribution-shifted tasks, where the downstream task diverges significantly from the pretraining domain, making both semantic alignment and the injection of novel task-specific information essential; and (ii) few-shot scenarios, where limited labeled examples demand that prompts focus attention on informative regions and introduce meaningful biases to enable rapid adaptation.

For distribution-shifted tasks, we assess VPT's performance across the 19 classification tasks in the VTAB-1k benchmark [73], which includes Natural (photographic), Specialized (e.g., medical or satellite), and Structured (geometric reasoning) categories. Following prior work [33, 42], we order the tasks by similarity to the pretraining distribution (ImageNet [10]), using the performance ratio between linear probing and full fine-tuning (LP/FT) as a proxy: larger ratios indicate stronger alignment, while smaller ratios suggest greater divergence. Figure 1(a) shows the resulting task ordering, from most dissimilar (dSprites/loc) to most similar (Sun397).

Figure 1(b) reports the relative performance of VPT compared to full fine-tuning, measured as $(\text{Acc}_{\text{VPT}} - \text{Acc}_{\text{full}})/\text{Acc}_{\text{full}}$, across the ordered tasks. The results show that VPT performs strongly on the top 30% most similar tasks (averaging +7.6%) but suffers sharp performance degradation on the bottom 30% most dissimilar tasks (averaging -34.6%), where greater representational shifts are required. This highlights VPT's limited adaptability under distribution shift.

In the few-shot setting, we compare VPT's performance with 8 training samples per class against full-data VPT on the CUB-200-2011 dataset [59], using both supervised [13] and MoCo-v3 [7] self-supervised pretrained backbones. As shown in Figure 1(c), few-shot VPT underperforms its full-data counterpart in both cases: the supervised backbone incurs a modest drop of 9.5%, while the self-supervised backbone suffers a much larger drop of 40.72%. This pronounced decline highlights the challenge of few-shot adaptation, especially for self-supervised models.

Together, these results reveal that VPT struggles to adapt in settings where its core mechanisms–focused attention and representational enrichment–are most essential. In the following section, we quantitatively analyze VPT's behavior with respect to semantic token selection and bias injection, and demonstrate that its failures in these challenging scenarios stem from underutilization of these mechanisms. This motivates a closer investigation into how to better activate and enhance these roles.

**Prompts Fail to Specialize**    To evaluate whether prompts attend to meaningful input tokens, we examine the attention matrix $\mathbf{S}_{\mathbf{P}_0 \mathbf{X}_0} \in \mathbb{R}^{N_p \times N_e}$ from (3), which encodes the attention weights from each prompt to the input tokens. Since identifying truly semantic tokens would require additional supervision (e.g., segmentation labels), we instead use the concentration of attention as a proxy for

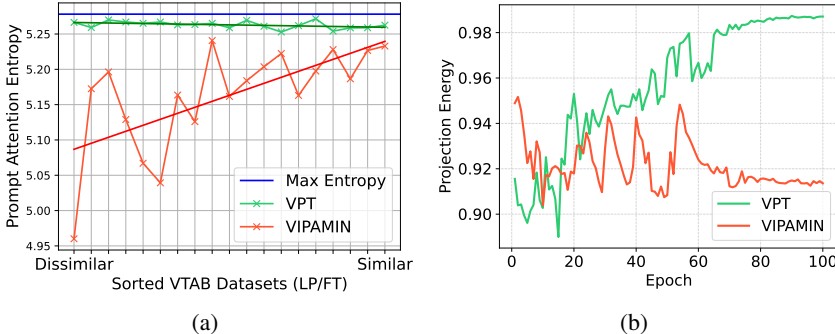

(a)                                                        (b)

Figure 2: **Failure modes of VPT.** (a) Prompt attention entropy of the fully trained model for each VTAB dataset, sorted by its LP ratio. The topmost blue line represents the maximum prompt attention entropy attainable, $\ln(N_e)$. (b) Subspace projection energy of $\mathbf{P}_0\mathbf{W}_V$ onto $\mathrm{SA}(\mathbf{X}_0)$, tracked during training on dSprites/loc (the most dissimilar task). Both figures are the training results of MoCo-v3 pretrained model.

specialization. For each prompt $\mathbf{p}_i$, we compute the entropy of its normalized attention distribution:

$$H(\mathbf{p}_i) = -\sum_{j=1}^{N_e} \frac{(\mathbf{S}_{\mathbf{P}_0\mathbf{X}_0})_{ij}}{S_i} \ln\left(\frac{(\mathbf{S}_{\mathbf{P}_0\mathbf{X}_0})_{ij}}{S_i}\right), \quad S_i = \sum_{k=1}^{N_e} (\mathbf{S}_{\mathbf{P}_0\mathbf{X}_0})_{ik} \tag{5}$$

where $(\mathbf{S}_{\mathbf{P}_0\mathbf{X}_0})_{ij} \propto \exp\left(\mathbf{p}_i^\top \mathbf{W}_Q \mathbf{W}_K^\top \mathbf{x}_j / \sqrt{d}\right)$. This quantity is upper bounded by $\ln(N_e)$; lower values indicate more concentrated (i.e., specialized) attention.

Figure 2(a) reports the average prompt attention entropy across all prompts for each VTAB-1k task, computed from 256 randomly selected training images and extracted from the final ViT layer. We observe that VPT consistently yields near-maximal entropy across tasks, regardless of their similarity to the pretraining distribution. This indicates that VPT fails to specialize prompt attention and instead distributes focus uniformly over all input tokens. While such broad attention may suffice for tasks closely aligned with the pretraining domain, it limits adaptability in more dissimilar tasks, where selective attention is needed to suppress irrelevant features and amplify task-specific signals.

**Prompts Fail to Inject Novel Semantic Bias**    To evaluate whether prompts introduce novel representational directions into the embedding space, we examine the relationship between $\mathbf{P}_0\mathbf{W}_V$ and $\mathrm{SA}(\mathbf{X}_0)$. As indicated in (4), $\mathbf{P}_0\mathbf{W}_V$ contributes a prompt-dependent bias to the self-attention output. However, if the row space of $\mathbf{P}_0\mathbf{W}_V$ is contained within that of $\mathrm{SA}(\mathbf{X}_0)$–i.e., $\mathrm{span}((\mathbf{P}_0\mathbf{W}_V)^\top) \subseteq \mathrm{span}(\mathrm{SA}(\mathbf{X}_0)^\top)$–then prompts fail to expand the representation space beyond what is already captured by the frozen backbone. To quantify this, we compute the *projection energy*, which measures the fraction of prompt signal contained within the pretrained self-attention subspace. Given $\mathbf{A} = (\mathbf{P}_0\mathbf{W}_V)^\top$ and $\mathbf{B} = (\mathrm{SA}(\mathbf{X}_0))^\top$, we define:

$$\mathrm{ProjectionEnergy}(\mathbf{A} \rightarrow \mathbf{B}) = \frac{\|\mathbf{P}_B\mathbf{A}\|_F^2}{\|\mathbf{A}\|_F^2}, \text{ where } \mathbf{P}_B = \mathbf{B}(\mathbf{B}^\top\mathbf{B})^{-1}\mathbf{B}^\top. \tag{6}$$

A value near 1 indicates that prompts contribute little to new representational directions, while a lower value suggests successful injection of orthogonal information. Figure 2(b) plots the projection energy of $\mathbf{P}_0\mathbf{W}_V$ onto $\mathrm{SA}(\mathbf{X}_0)$ during training on dSprites/loc–the most dissimilar task in VTAB-1k. VPT converges to a projection energy near 1, revealing that the prompt output collapses into the frozen subspace and fails to provide novel signals needed for adaptation.

Together, these findings show that VPT fails to specialize attention or introduce task-specific semantic bias–capabilities that are particularly critical for distribution-shifted and few-shot settings. To overcome these limitations, we introduce VIPAMIN: **V**isual **I**nitialization of **P**rompts via **A**ttention-guided **M**atching and **I**njection of **N**ovelty. VIPAMIN initializes prompts by aligning them with semantically relevant input embeddings while injecting directions orthogonal to the frozen subspace. As shown in Figures 2(a) and 2(b) (red curves), VIPAMIN produces lower attention entropy and projection energy on dissimilar tasks, enabling more effective adaptation in challenging scenarios.

# 4   Methodology: VIPAMIN

We propose VIPAMIN, a novel initialization method for visual prompt tuning that enables prompts to (1) specialize their attention and (2) inject novel representational directions beyond the pretrained subspace. VIPAMIN consists of two complementary modules: **Matching module**, which addresses the issue of uniform attention by aligning each prompt with semantically coherent input tokens from the downstream task, thereby promoting specialization over meaningful local regions; and the **Orthogonalizing module**, which prevents representational collapse by projecting prompts away from the pretrained embedding subspace, facilitating the injection of task-specific information.

**Matching Module: Prompt Specialization via Semantically Coherent Token Matching**    The matching module encourages each prompt to focus on semantically coherent local regions from its initialization, with controllable locality. A key challenge is identifying such coherent regions without external supervision (e.g., segmentation labels). To address this, we leverage a pretrained ViT and extract the embedding matrix $\mathbf{E}_0^{\text{batch}} \in \mathbb{R}^{B \times (N_p + N_e) \times d}$ by passing $B$ downstream training images through the frozen model. After mean pooling across the batch, we obtain $\mathbf{E}_0 \in \mathbb{R}^{N_e \times d}$.

Next, for each randomly sampled prompt $\mathbf{p}_i \in \mathbb{R}^d$ (initialized using Xavier uniform [17]), we project both the prompt and the mean-pooled input embeddings into the key subspace $\mathbf{W}_K \in \mathbb{R}^{d \times d}$ of the first transformer block. We then compute cosine similarity between the projected prompt and each input token, and select the top-$k$ most aligned tokens:

$$\{\alpha_j\}_{j=1}^k = \texttt{TopKIndices}(\cos(\mathbf{p}_i^\top \mathbf{W}_K, \mathbf{E}_0 \mathbf{W}_K), k) \tag{7}$$

where $\texttt{TopKIndices}(\cdot, k)$ returns the indices of the top-$k$ values. In ViT-B/16, each token covers only 0.5% of the image, so semantically coherent regions typically span multiple tokens, which tend to cluster in the key space. This makes it likely that the top-$k$ tokens share similar semantics. Additional in-depth discussion is available in Appendix I.3.

Figure 3 shows an example where two randomly sampled prompts select distinct token groups. Each group focuses on different semantically coherent components of the image (e.g., display racks, top/bottom shelves of bread), confirming that random prompts select diverse yet meaningful regions.

The initialization vector for each prompt is obtained by averaging its matched token embeddings:

$$\mathbf{p}_i^{\text{avg}} \leftarrow \frac{1}{k} \sum_{j=1}^{k} (\mathbf{E}_0)_{\alpha_j}. \tag{8}$$

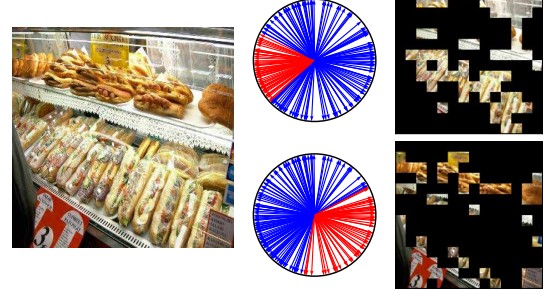

This encourages each prompt to focus on a coherent semantic region. The hyperparameter $k$ controls the locality of this focus. With a sufficient number of prompts ($N_p$), their union ensures broad coverage of salient image regions. This matching-based initialization is fully unsupervised and requires no additional annotation.

Figure 3: **Coherence of embeddings in key space.** For an image of a bakery, we show matched tokens for two randomly sampled prompts by highlighting the top-$k$ indices ($k = 60$) in red.

**Orthogonalizing Module: Injecting Novel Representational Bias**    While the matching module initializes prompts by directly leveraging existing semantic clusters in the embedding space, this alone may be insufficient for tasks that diverge significantly from the pretraining distribution. In such cases, prompts initialized solely from existing embeddings may cause the self-attention output in (4) to collapse into the subspace of $\text{SA}(\mathbf{E}_0)$, limiting representational diversity. To mitigate this, we introduce an orthogonal component into each prompt that lies outside the row space of $\text{SA}(\mathbf{E}_0)$.

Specifically, we compute the SVD of $\text{SA}(\mathbf{E}_0) = \mathbf{U}\mathbf{\Sigma}\mathbf{V}^\top$, where $\mathbf{V}$ spans the row subspace of $\text{SA}(\mathbf{E}_0)$. We then project a random prompt $\mathbf{p}_i$ through the value matrix $\mathbf{W}_V$, remove its projection onto the subspace, and reverse-map it back via the pseudoinverse of $\mathbf{W}_V$:

$$\mathbf{p}_i^{\text{orth}} \leftarrow (\mathbf{I} - \mathbf{V}\mathbf{V}^\top)(\mathbf{p}_i \mathbf{W}_V)(\mathbf{W}_V)^\dagger. \tag{9}$$

Table 1: Fine-tuning results for VTAB-1k benchmarks with pretrained **ViT-B/16**. The best and second-best results are highlighted in **bold** and underline, resp. Per-task results available in Appendix F.

| **MoCo-v3 pretrained ViT-B/16** | | | | | **MAE pretrained ViT-B/16** | | | | |
|---|---|---|---|---|---|---|---|---|---|
| Method | Natural | Specialized | Structured | Mean | Method | Natural | Specialized | Structured | Mean |
| Full | 71.95 | **84.72** | 51.98 | 66.23 | Full | 59.31 | 79.68 | 53.82 | 61.28 |
| VPT | 67.34 | 82.26 | 37.55 | 57.94 | VPT | 39.96 | 69.65 | 27.50 | 40.96 |
| GateVPT | 74.84 | 83.38 | 49.10 | 65.80 | GateVPT | 47.61 | 76.86 | 36.80 | 49.22 |
| SPT | 74.47 | 83.93 | 55.16 | 68.33 | SPT | 62.53 | **80.90** | 53.46 | 62.58 |
| **VIPAMIN** | **76.75** | 84.14 | **56.68** | **69.86** | **VIPAMIN** | **62.60** | 79.96 | **57.47** | **64.09** |

Each final prompt is a weighted combination of the matched (semantic) and orthogonal components:

$$\mathbf{p}_i^{\text{VIPAMIN}} \leftarrow (1 - \lambda)\mathbf{p}_i^{\text{avg}} + \lambda\mathbf{p}_i^{\text{orth}}, \tag{10}$$

where $\lambda \in [0, 1]$ controls the strength of orthogonalization. This hybrid strategy enables prompts to both specialize on meaningful regions and expand into new representational directions. Empirically, higher $\lambda$ values are beneficial for tasks that are semantically distant from the pretraining domain.

## 5 Experiments

### 5.1 Experiment Setup

**Datasets and Models**   We evaluate our method on two image classification benchmarks: Visual Task Adaptation Benchmark (VTAB-1k) [73] and Fine-Grained Visual Categorization (FGVC). To assess performance under varying degrees of distribution shift from pretraining, we use VTAB-1k, which comprises 19 visual classification tasks grouped into three categories: *Natural*; consisting of everyday images captured with standard cameras, *Specialized*; containing images taken using domain-specific equipment, and *Structured*; involving tasks that require geometric understanding. For few-shot evaluation, we construct $k$-shot settings by sampling the training set of five FGVC datasets: CUB-200-2011 [59], NABirds [56], Stanford Dogs [30], Stanford Cars [15], and Oxford Flowers [46]. While these datasets are relatively easier under full supervision compared to VTAB-1k, they effectively reveal the limitations of self-supervised VPT in data-scarce regimes, as illustrated in Figure 1(c). Additional benchmark details are provided in Appendix D.

We use ViT-B/16 as the backbone model unless otherwise specified. Our method is evaluated using two widely adopted self-supervised pretrained backbones: Momentum Contrast v3 (MoCo-v3) [7], and Masked Autoencoders (MAE) [21]. More experimental details are reported in Appendix D.

**Baselines**   We use VPT [27] and SPT [66] as our main baselines. VPT initializes prompts randomly using Xavier uniform initialization [18]. SPT performs $K$-means clustering on reshaped input tokens $\mathbf{E}_0' \in \mathbb{R}^{BN_e \times d}$ extracted from $\mathbf{E}_0 \in \mathbb{R}^{B \times N_e \times d}$, and uses the resulting $N_p$ centroids as prompt initializations. Due to the high computational cost of clustering in SPT (e.g., ∼27 days on CUB-200-2011), we instead adopt a low-cost variant that randomly samples $N_p$ tokens from $\mathbf{E}_0'$, which matches SPT's reported accuracy (Table 4(b) in [66]). We refer to this variant as **SPT/rand**. For SPT/rand, we use the full training set to obtain $\mathbf{E}_0$, while VIPAMIN uses only a random mini-batch of size 256. In the FGVC few-shot experiment, both SPT/rand and VIPAMIN sample their respective batches from the combined training set and validation set.

### 5.2 Main Result

Table 1 presents the performance of VIPAMIN, baseline visual prompt tuning methods, and full fine-tuning (Full) across the 19 VTAB-1k tasks, grouped into Natural, Specialized, and Structured categories. Across both MoCo-v3 and MAE pretrained backbones, VIPAMIN achieves the highest average accuracy. Notably, it delivers substantial improvements on Structured tasks, outperforming VPT by +19.13 % (MoCo-v3) and +29.97 % (MAE), demonstrating its effectiveness in tasks requiring spatial or relational reasoning. Importantly, these gains do not come at the expense of performance on simpler tasks: VIPAMIN also achieves the best or comparable results on the Natural category under both backbones, confirming its broad generalization. For MAE, VIPAMIN is the first prompt-based

Table 2: Few-shot accuracy (%) for $k$-shot classification across five FGVC datasets averaged over three independent runs; the final column in each block is the mean accuracy of the five datasets.

| Method | $k=1$ | | | | | | $k=2$ | | | | | | $k=4$ | | | | | | $k=8$ | | | | | |
|---|---|---|---|---|---|---|---|---|---|---|---|---|---|---|---|---|---|---|---|---|---|---|---|---|
| | CUB | Birds | Flowers | Dogs | Cars | Mean | CUB | Birds | Flowers | Dogs | Cars | Mean | CUB | Birds | Flowers | Dogs | Cars | Mean | CUB | Birds | Flowers | Dogs | Cars | Mean |
| VPT | 15.7 | 7.7 | 31.4 | 31.2 | 4.7 | 18.1 | 15.6 | 11.7 | 59.0 | 45.4 | 6.4 | 27.6 | 31.4 | 14.3 | 66.2 | 36.8 | 9.9 | 31.7 | 37.3 | 17.2 | 77.8 | 62.8 | 13.5 | 41.7 |
| SPT/rand | 17.2 | 11.7 | 48.9 | 35.5 | 5.3 | 23.7 | 29.8 | 22.5 | 70.4 | 49.0 | 10.9 | 36.5 | 51.7 | 40.5 | 84.6 | 59.8 | 21.7 | 51.7 | 66.6 | 55.0 | 92.9 | 69.1 | 43.8 | 65.5 |
| **VIPAMIN** | **20.1** | **12.6** | **52.8** | **37.5** | **5.7** | **25.8** | **36.0** | **23.1** | **71.6** | **49.4** | **11.1** | **38.2** | **53.7** | **41.0** | **85.1** | **60.3** | **21.9** | **52.4** | **68.6** | **55.1** | **94.3** | **70.0** | **43.8** | **66.4** |

method to surpass full fine-tuning across all VTAB-1k categories. While it performs strongly on MoCo-v3 as well, it falls slightly short in the Specialized group.

Table 2 presents $k$-shot classification results on five fine-grained benchmarks for $k \in \{1, 2, 4, 8\}$. VIPAMIN consistently achieves the best performance across all settings. In terms of mean accuracy, VIPAMIN outperforms VPT by 7.7% at $k = 1$, and the margin grows to 24.7% at $k = 8$. While the gains over the SPT/rand baseline are more modest (1–2%), they remain consistent, underscoring VIPAMIN's superior adaptability in data-scarce regimes.

## 5.3 Scalability Analysis

**Extension to VPT-Deep** While our primary analysis has focused on the non-intrusive VPT-Shallow–where prompts are prepended before the first transformer block and propagates–we also evaluate our main ideas on VPT-Deep variant. In VPT-Deep, a separate prompt $\mathbf{P}_l$ is prepended to the token sequence at each block: $B_{l+1}([\mathbf{P}_l; \mathbf{X}_l])$ for $l \in [L-1]$. Since each prompt operates only within its corresponding block, the token-selection dynamics that are critical in VPT-Shallow are less prominent. However, we hypothesize that prompt-subspace collapse can still occur independently within each block's local prompt space (see Appendix I.5 for further discussion).

In VIPAMIN-Deep, we apply the modules to each block's input $\mathbf{X}_l$, using a fixed prompt length of 20. We compare VIPAMIN-Deep against established PEFT methods such as Adapter [4] and Bias Tuning [52], as well as

Table 3: Fine-tuning results for VTAB-1k benchmarks with **MoCo-v3** pretrained ViT-B/16. Specialized tasks are abbreviated as *Spec*.

| Method | | Natural | Spec | Structured | Mean |
|---|---|---|---|---|---|
| Full | | 71.95 | 84.72 | 51.98 | 66.23 |
| Linear Probing [24] | | 67.46 | 81.08 | 30.33 | 54.69 |
| Partial-1 [71] | | 72.31 | 84.58 | 47.89 | 64.61 |
| Bias [52] | | 72.89 | 81.14 | 53.43 | 66.43 |
| Adapter [4] | | 74.19 | 82.66 | 47.69 | 64.82 |
| VPT-Deep [27] | | 70.27 | 83.04 | 42.38 | 61.22 |
| E$^2$VPT [20] | | 76.47 | **87.28** | 54.91 | 69.67 |
| SPT-Deep [66] | | 76.20 | 84.95 | 58.36 | 70.53 |
| iVPT [76] | | 76.12 | 84.51 | 57.88 | 70.21 |
| DA-VPT [53] | | 74.24 | 83.21 | 55.23 | 69.13 |
| VFPT [72] | | 77.47 | 85.76 | 58.74 | **71.33** |
| **VIPAMIN-Deep** | | 77.68 | 84.79 | **58.80** | 71.23 |

more intrusive approaches that require architectural changes, including E$^2$VPT [20] and iVPT [76] (see Appendix C.1 for a detailed breakdown of the overhead). As shown in Table 3, VIPAMIN-Deep outperforms full fine-tuning across all three VTAB-1k categories, demonstrating strong adaptability. It also surpasses the best-known prompt initialization baseline, SPT-Deep. While VFPT slightly outperforms VIPAMIN-Deep in mean accuracy, our method achieves competitive performance without architectural modifications, showing its simplicity and generality.

**Scaling Model and Prompt Length** A key limitation of VPT is its reduced performance with larger backbones, where increased complexity hinders optimization [27]. A similar trend appears with longer prompts, where added tokens often fail to improve representation [31]. As shown in Fig. 4 (a, b), VIPAMIN mitigates both issues, maintaining stable convergence even with ViT-H/14, where VPT fails. Furthermore, VIPAMIN is the only method that consistently benefits from increased prompt length, with the performance gap widening as prompt capacity grows. This scalability advantage stems from its semantic bias injection, which enables more effective use of additional prompts.

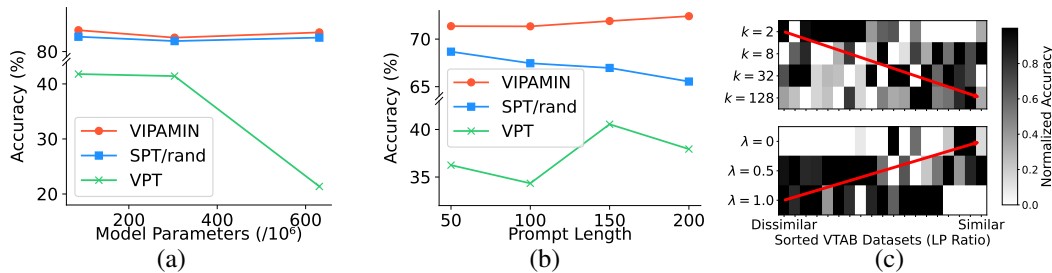

Figure 4: (a) MAE-pretrained ViT-B/L/H are evaluated on Oxford Flowers102 from FGVC. (b) We assess the effect of prompt length in MoCo-v3 on CIFAR-100. (c) We evaluate the performance of VIPAMIN on MAE across VTAB-1k, varying the hyperparameters $k$ and $\lambda$. Datasets are ordered by LP ratio from lowest to highest. We select the learning rate that yields the highest validation accuracy and normalize the resulting top-1 accuracy to the $[0, 1]$ range for each dataset (column-wise). A rough trend of the best-performing hyperparameter combinations is indicated by the red arrow.

## 5.4 Ablation Study and Further Analysis

**Role of $k$ and $\lambda$** VIPAMIN introduces two key hyperparameters: $k$ in (7), which specifies the number of tokens to which each prompt attends, and $\lambda$ in (10), which controls the degree of orthogonal bias injected into the prompt initialization. Figure 4 (c) clarifies the roles of $k$ and $\lambda$. For tasks that exhibit significant distributional shift, smaller $k$ and larger $\lambda$ yield the best performance. This suggests that such tasks benefit from more localized attention and stronger orthogonalization, which together enable the model to focus on task-specific features not captured by the pretrained backbone. In contrast, tasks more closely aligned with the pretraining distribution tend to favor larger $k$ and smaller $\lambda$, indicating that broader attention and greater reliance on pretrained representations are advantageous.

**Ablation on VIPAMIN Modules** We conduct an ablation study to evaluate the contributions of the two core components of VIPAMIN: the matching module and the orthogonalizing module. The results are summarized in Table 4. Incorporating the matching module alone already yields notable improvements over the SPT baseline in the Natural group, indicating that semantic token selection is particularly beneficial for tasks that are well aligned to the pretraining domain. In contrast, tasks in the Specialized group

Table 4: Ablation study on the two modules in VIPAMIN, using MoCo-v3 pretrained models evaluated on the VTAB-1k benchmark.

| Matching | Orth | Natural | Spec | Structured |
|:---:|:---:|:---:|:---:|:---:|
| *SPT baseline* | | *74.47* | *83.93* | *55.16* |
| ✓ | | 76.50 | 82.85 | 56.51 |
| ✓ | ✓ | **76.75** | **84.14** | **56.68** |

often require the model to acquire novel semantic representations (e.g., identifying abnormal retinal features). Here, the orthogonalizing module proves essential by projecting the prompt away from the pretrained embedding space, thereby enabling the injection of new, domain-specific information.

**Impact of Orthogonalizing Module on Task-Aligned Representation Learning** While the orthogonalizing module ensures that the generated prompt lies outside the pretrained subspace, it remains an open question whether this nudging induces representations aligned with task-relevant directions that ultimately improve classification performance. To empirically examine this, we analyze the Last-Layer CLS Representations (LLCR)–i.e., the output embeddings of the final transformer block prior to the classification head. As a reference, we treat the LLCR obtained from a fully fine-tuned model as the oracle, denoted by $\text{LLCR}_{\text{ft}}$. We then compare this to $\text{LLCR}_{\text{no-orth}}$ and $\text{LLCR}_{\text{VIPAMIN}}$, which are obtained from models trained using only the matching module and both the matching and orthogonalizing modules, respectively. We focus on the Diabetic Retinopathy dataset in VTAB-1k, given its clinical nature and the demand for novel domain-specific knowledge. Our evaluation is two-fold: instance-level analysis and subspace-level analysis. First, at the instance level, we compute the cosine distance between $\text{LLCR}_{\text{ft}}$ and each of the compared methods for every validation sample. A Wilcoxon signed-rank test [68] on the paired distances $d_{\cos}(\text{LLCR}_{\text{ft}}, \text{LLCR}_{\text{no-orth}}) - d_{\cos}(\text{LLCR}_{\text{ft}}, \text{LLCR}_{\text{VIPAMIN}})$ yields a $p$-value $< 2 \times 10^{-7}$, indicating

that $\text{LLCR}_{\text{VIPAMIN}}$ is closer to the fine-tuned oracle than $\text{LLCR}_{\text{no-orth}}$. Second, at the subspace level, we assess representational similarity by viewing the span of each LLCR set as a subspace and computing the Grassmannian distance [39] to $\text{LLCR}_{\text{ft}}$. The distance between $\text{LLCR}_{\text{ft}}$ and $\text{LLCR}_{\text{no-orth}}$ is 14.56, whereas the distance between $\text{LLCR}_{\text{ft}}$ and $\text{LLCR}_{\text{VIPAMIN}}$ is reduced to 14.03. These findings suggest that the orthogonalizing module not only increases representational diversity but also actively guides the model toward task-aligned subspaces—thereby nudging the representations in a direction that approximates the fully fine-tuned oracle more effectively.

**Grad-CAM Inspection** We analyze Grad-CAM [54] visualizations for two datasets in VTAB-1k: dSprites/location and Sun397, corresponding to the lowest (most dissimilar to pretraining) and highest (most similar) LP ratios, respectively. The results are shown in Figure 5. In the case of VPT, we observe a uniform attention pattern, with the model broadly attending to most image patches, regardless of the dataset. In contrast, SPT/rand produces more localized attention but often fails to highlight semantically critical regions. For VIPAMIN, the effect of the hyperparameter $k$ is clearly visible when comparing $k = 2$ and $k = 128$. A higher $k$ leads to broader attention across the image, while a lower $k$ concentrates attention on a narrower region. Consequently, for tasks that require localized attention (e.g., dSprites/location), setting a smaller $k$ is advantageous.

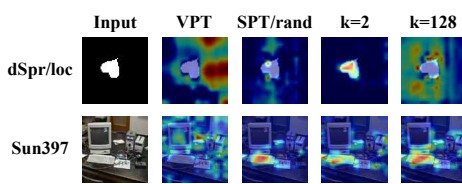

Figure 5: Red regions indicate their critical role in the model's class prediction. The cases of $k = 2$ and $k = 128$ correspond to VIPAMIN models trained with their respective hyperparameter, $k$.

# 6 Conclusion

We introduced VIPAMIN, a parameter-free initialization scheme for visual prompts in self-supervised models that jointly aligns prompts with semantically meaningful regions and injects novel representational directions orthogonal to the frozen embedding space of vision transformers. Unlike prior approaches that rely on auxiliary modules or architectural modifications, VIPAMIN requires only two lightweight matrix operations, introduces no training overhead, and integrates seamlessly into existing pipelines. Extensive experiments across 24 vision tasks, including data-scarce settings, demonstrate consistent performance improvements.

# Acknowledgements

This work was supported by the National Research Foundation of Korea (NRF) grant funded by the Korea government (MSIT) (No. RS-2024-00408003, RS-2025-00516153 and RS-2024-00444862).

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

# Appendix Overview

The appendix section includes additional experimental results and discussions of our NeurIPS 2025 submission: *VIPAMIN: Visual Prompt Initialization via Embedding Selection and Subspace Expansion*, summarized as follows:

- Section A discusses limitations and broader impact
- Section B reviews related works in depth
- Section C presents the VIPAMIN algorithm and an overhead analysis
- Section D elaborates experimental setup
- Section E justifies the use of SPT/rand baseline
- Section F reports full VTAB-1k results
- Section G reports full FGVC few-shot results
- Section H studies adaptability of VIPAMIN, such as generalization across models, scales, and modality
- Section I provides further analyses, including GradCAM visualization, feature-selection ablations, hyperparameter roles, and analysis on supervised VPT

## A   Limitations and Broader Impact

**Limitations**   While our method demonstrates significant improvements for self-supervised visual prompt tuning (VPT), several important questions remain open for future investigation. (1) Can the principles behind VIPAMIN generalize to other modalities, such as text or multimodal settings? (2) What are the fundamental representational differences between supervised and self-supervised pretrained models, and how do these differences affect prompt behavior? (3) Is there a principled approach for selecting the hyperparameters introduced in VIPAMIN (e.g., $k$, $\lambda$), to reduce reliance on empirical tuning? Addressing these questions would enhance the applicability and theoretical understanding of prompt-based adaptation in broader contexts.

**Broader Impact**   This work contributes to the advancement of parameter-efficient transfer learning by improving the reliability and adaptability of visual prompt tuning, particularly in self-supervised settings. By addressing key failure modes–namely, uniform attention and representational collapse– our method enhances the utility of VPT under distribution shift and in data-constrained regimes. These improvements may facilitate broader adoption of foundation models in domains where full fine-tuning is computationally prohibitive or impractical.

More broadly, the VPT paradigm represents a shift toward more sustainable and accessible AI. By decoupling task adaptation from full model retraining, prompt tuning can significantly reduce the computational and environmental costs associated with deploying large-scale vision models. This has the potential to democratize the use of powerful pretrained models, enabling resource-constrained institutions to more effectively leverage state-of-the-art vision systems.

## B   Related Works

### B.1   Detailed Review of Related Works

**Theoretical Foundations of Prompt Tuning**   While prompt tuning has shown strong empirical performance, its underlying mechanisms remain incompletely understood. Recent theoretical work suggests that, under simplified conditions, gradient descent on prompt tokens can selectively modulate attention–amplifying focus on label-relevant tokens while suppressing noise [48]. Other studies underscore the expressive limitations of prompt-based methods. For example, [51, 62] argue that methods like prefix tuning introduce low-rank output biases without modifying attention patterns among input tokens, largely retrieving existing knowledge rather than enriching representations.

Our study extends these analyses by empirically examining self-supervised visual transformers under realistic training conditions. We find that randomly initialized prompts, commonly used in VPT, often fail to specialize or retrieve meaningful semantics during optimization. Moreover, the induced

low-rank bias frequently collapses into the subspace of the frozen self-attention output–suggesting that, in practice, prompt signals may be absorbed into the pretrained representation space, limiting their potential to guide adaptation.

**Prompt Initialization in Language and Vision**    Prompt tuning originated from hard prompts–manually crafted natural language inputs–to steer pretrained language models [37]. As the field shifted to continuous, trainable soft prompts, concerns arose around their optimization, particularly due to slow convergence under standard gradient descent [55]. This motivated prompt initialization strategies aimed at improving stability and performance. An early study by [36] compared Xavier-uniform initialization [17] to alternatives using frequent word embeddings and label-token embeddings. While these improved performance in smaller models, gains diminished with model scale. More recently, Pretrained Prompt Tuning (PPT) [19] showed that soft prompts perform well with ample data but struggle in few-shot settings due to poor initialization. PPT addresses this by pretraining prompts while keeping the backbone frozen, yielding strong improvements in data-scarce regimes.

Despite progress in the language domain, prompt initialization techniques have not been thoroughly explored in the vision domain. To our knowledge, the only prior work explicitly addressing visual prompt initialization is Self-Prompt Tuning (SPT) [66]. SPT observes that, during training, prompt representations tend to converge to the distribution of patch embeddings. Motivated by this, SPT proposes initializing prompts using prototype embeddings derived from clustering patch tokens, thereby accelerating convergence.

In contrast to SPT, our work takes a fundamentally different approach. Rather than relying solely on clustering-based prototypes, we identify and address the two core limitations of visual prompt tuning–uniform attention and subspace collapse. Our proposed method, VIPAMIN, not only facilitates fast convergence and stable training through principled initialization but also enhances the representational diversity of prompts, enabling better adaptation under challenging conditions.

**Prompt Tuning in Few-Shot Learning**    Few-shot learning is a natural use case for prompt tuning, especially in resource-constrained settings. While most prior studies focus on NLP and multimodal domains [19, 35, 62, 77], they consistently report limitations in low-data regimes. PPT [19] was among the first to identify poor performance in few-shot scenarios, attributing it to optimization challenges, and proposed a prompt-only pretraining strategy to improve generalization.

In the vision domain, [75] similarly highlight the shortcomings of visual prompt tuning in extreme few-shot settings (e.g., $k = 1, 2$), and introduce NOAH–a neural architecture search framework that integrates multiple parameter-efficient tuning (PEFT) methods to enhance few-shot performance.

## B.2    In-Depth Comparison with Related Works

**Self-Prompt Tuning (SPT)**    The underlying intuition behind Self-Prompt Tuning (SPT) is that the convergence of the prompt could be improved by initializing prompts using prototypical representations of patch tokens. However, this approach does not involve token-level selection; instead, it constructs prototypes without explicitly identifying which tokens carry semantically meaningful information. In contrast, VIPAMIN performs token-level selection in a scalable way and further enhances adaptation by injecting novel, orthogonal directions–an inductive bias that directly contrasts with SPT's prototype-based initialization. This combination proves more effective, especially under data-scarce scenarios.

**Gated Prompt Tuning**    In [70], the authors highlight that the expression of pretrained knowledge differs significantly between supervised and self-supervised models. Specifically, through the lens of deep image priors, they observe that the retention of image-specific information varies across transformer blocks, and the distribution of task-relevant features depends on the pretraining strategy. Our work complements these findings in two key ways. First, we identify and analyze specific failure modes of VPT in self-supervised settings, offering a clearer view of how prompts interact with frozen embeddings. Second, we move beyond characterizing these differences to propose a practical method, VIPAMIN, that actively exploits the structural properties of self-supervised representations to improve adaptation.

**SPARC (Continual Prompt Learning)**   Recent work in continual learning with prompts has proposed initializing new prompts in the orthogonal subspace of previously learned ones, thereby facilitating adaptation to novel tasks while preserving prior knowledge. This approach, exemplified by SPARC, seeks to mitigate catastrophic forgetting by reusing task-specific prompts and conditionally initializing new prompts such that $\langle \mathbf{P}_{new}, \mathbf{P}_{old} \rangle$ [26]. While this strategy bears similarity to our orthogonal initialization, the focus of our method is fundamentally different. Rather than emphasizing prompt reuse across tasks, we investigate the geometric relationship between prompt tokens and the representation space, and how this interaction evolves during training. In particular, we formally motivate the benefit of initializing prompts in the orthogonal complement of the dominant subspace of representations, highlighting its potential to enhance adaptation by directing attention toward underutilized directions.

**Token-Coordinated Prompt Attention (TCPA)**   TCPA [41] addresses the limitation of shared prompts in VPT by assigning token-specific prompts and modulating attention separately for CLS and patch tokens. This is achieved by explicitly masking and reweighting attention patterns throughout training. While VIPAMIN shares the motivation of improving prompt-token interaction, it takes a different route: instead of modifying attention dynamics during training, it initializes prompts using token clusters from the pretrained key space, ensuring semantic alignment from the outset. More importantly, VIPAMIN explicitly mitigates prompt collapse into the representation subspace via orthogonalization. As shown in Table 4 and Figure S3, this proves particularly advantageous for dissimilar and few-shot tasks where prompt diversity and representational novelty are essential.

## C   Algorithm

Detailed algorithm description for VIPAMIN is introduced in Algorithm 1.

---

**Algorithm 1** VIPAMIN

---

**Input:** Minibatch $\mathcal{B}$, Pretrained Vision Transformer $\mathcal{M}$ with parameter $\theta^{pre}$, Locality factor of the matching module $k$, Relative strength of the orthogonalizing module $\lambda$, Prompt length $N_p$, Randomly initialized prompts $\mathbf{P}_0^{rand} = \{p_1^{rand}, p_2^{rand}, \cdots, p_{N_p}^{rand}\}$
**Output:** Set of generated prompts $\mathbf{P}_0 = \{p_1, p_2, \cdots, p_{N_p}\}$
**Input Preparation**
Fetch self-attention weights of the first block, $\{\mathbf{W}_Q, \mathbf{W}_K, \mathbf{W}_V, \mathbf{b}_Q, \mathbf{b}_K, \mathbf{b}_V\} \subset \theta^{pre1}$
Fetch $\mathbf{E}_0 := \texttt{PosEmbed} + \texttt{PatchEmbed}(\mathcal{B})$ via feedforwarding $\mathcal{B}$ through $\mathcal{M}$
Fetch $\text{SA}(\mathbf{E}_0) \leftarrow \text{softmax}\left((\mathbf{E}_0\mathbf{W}_Q + \mathbf{b}_Q)(\mathbf{E}_0\mathbf{W}_K + \mathbf{b}_K)^T / \sqrt{d}\right)(\mathbf{E}_0\mathbf{W}_V + \mathbf{b}_W)$
**Matching Module**
$\text{IndexMap} = \texttt{RowWiseTopKIndices}(\overline{\mathbf{P}_0^{rand}\mathbf{W}_K} \times \overline{\mathbf{E}_0\mathbf{W}_K}^T, k)^2$
**for** $i \leftarrow 1$ **to** $N_p$ **do**
    $\mathbf{p}_i^{match} \leftarrow \texttt{RowWiseMean}(\mathbf{E}_0[\text{IndexMap}_i, :])$
**end for**
$\mathbf{P}_0^{match} \leftarrow (\mathbf{p}_0^{match}, \mathbf{p}_1^{match}, \cdots, \mathbf{p}_{N_p}^{match})^T$
**Orthogonalizing Module**
SVD Decompose $\text{SA}(\mathbf{E}_0), \mathbf{U}\Sigma\mathbf{V}^T = \text{SA}(\mathbf{E}_0)$
**for** $i \leftarrow 1$ **to** $N_p$ **do**
    $\mathbf{p}_i^{orth} \leftarrow (\mathbf{I} - \mathbf{V}\mathbf{V}^T)(\mathbf{p}_i^{rand}\mathbf{W}_V + \mathbf{b}_V)\mathbf{W}_V^{\dagger} - \mathbf{b}_V$
**end for**
$\mathbf{P}_0^{orth} \leftarrow (\mathbf{p}_0^{orth}, \mathbf{p}_1^{orth}, \cdots, \mathbf{p}_{N_p}^{orth})^T$
$\mathbf{P}_0 \leftarrow (1 - \lambda)\mathbf{P}_0^{match} + \lambda\mathbf{P}_0^{orth}$
Perform prompt tuning with prompt-prepended input, $[\mathbf{P}_0; \mathbf{E}_0]$

---

[1]For models which don't employ biases in the self-attention module, $\mathbf{b}_Q, \mathbf{b}_K, \mathbf{b}_V$ are set to zero vectors.
[2]$\overline{\mathbf{A}} := \text{diag}\left(\|\mathbf{A}_i\|_2^{-1}\right)\mathbf{A}$

Table S1: **Computational overhead analysis on Flower-102.** MoCo-v3 pretrained model is used to report the accuracy.

|  | Tuned/Total | Additional FLOPs | Init. GFLOPs | Accuracy |
|---|---|---|---|---|
| VPT | 0.30% | 0 (17.7 GFLOPs) | 0 | 90.2 |
| E$^2$VPT | 0.20% | < 0 | 0 | 91.3 |
| iVPT | 0.31% | < 1M | 0 | N/A |
| VFPT | 0.30% | 5.5M | 0 | 92.4 |
| SPT-Deep | 0.30% | 0 | 3300 | 93.2 |
| VIPAMIN-Deep | 0.30% | 0 | 1.73 | **94.0** |

## C.1 Overhead Analysis

The computational overhead introduced by our method is minimal. We execute each module 5 times using model ViT-B/16, prompt length of 100, and training batch from Resisc45. Excluding the time required for model and dataset loading (denoted as **Input Preparation** in Algorithm 1), the average wall-clock time to execute the core logic of the matching module is 3.94 seconds, with a standard deviation of 0.06 seconds. The orthogonalizing module incurs slightly greater overhead due to its reliance on singular value decomposition and pseudo-inverse computation; its core logic requires 9.14 seconds on average, with a standard deviation of 0.11 seconds. Importantly, this overhead is incurred only once during initialization and is subsequently amortized over the full training process.

Continuing the comparison with other VPT-Deep variants presented in Section 5.3, we further substantiate the superior efficiency of VIPAMIN-Deep in multiple dimensions. Table S1 shows the result. First, VIPAMIN, along with SPT-Deep, introduces no additional parameters to be tuned and no extra FLOPs relative to vanilla VPT. Furthermore, for one-time initialization, SPT incurs an overhead that is orders of magnitude larger than ours. In addition, the total additional runtime of VIPAMIN over VPT is under 30 seconds, and the memory overhead is modest–the peak GPU allocation is 2.2 GB on an NVIDIA A6000. Lastly, the seemingly superior efficiency of E$^2$VPT is largely due to its pruning module, which entails a two-stage training procedure. Overall, VIPAMIN-Deep delivers the strongest accuracy-compute trade-off among the VPT-Deep variants.

# D  Experimental Details

**Benchmark Datasets Specification**    In this section, we provide additional details on the benchmarks used for evaluation: VTAB-1k and FGVC. The 19 datasets included in VTAB-1k are summarized in Table S2. For the FGVC few-shot experiments, we sample from the training set and use the validation set for both hyperparameter tuning and extracting embeddings for the initialization procedures of SPT/rand and VIPAMIN. Additional benchmark specifications are also provided in Table S3.

**Hyperparameter Specification**    Our hyperparameter settings largely follow the configuration used in [66]. However, for self-supervised backbones, we narrow the learning rate range based on the recommendations from [70]. The full set of hyperparameters is detailed in Table S4.

**Hyperparameter Tuning and Evaluation Protocol**    We perform a grid search over a predefined hyperparameter pool on the training set and select the configuration that achieves the highest validation accuracy. The final reported performance corresponds to the average top-1 accuracy on the test set, computed over three independent runs using the selected hyperparameters. The complete tuning pool is provided in Table S4.

**Training Configurations**    We use the AdamW optimizer with a batch size of 32. The learning rate follows a cosine decay schedule with a linear warm-up period of 10 epochs, consistent with prior work [66]. Unless otherwise specified, we use a prompt length of 100.

**Reproducibility**    The implementation of VIPAMIN is based on PyTorch [50], with VTAB-1k datasets loaded via TensorFlow Datasets [1], following established practices in [20, 66, 70]. All experiments were conducted on NVIDIA A6000 GPUs with 40GB of memory.

Table S2: Specifications of the VTAB-1k [73] benchmark.

| Dataset | Description | # Classes | Train | Val | Test | Preprocessing |
|---|---|---|---|---|---|---|
| CIFAR-100 [32] | | 100 | | | 10,000 | |
| Caltech101 [14] | | 102 | | | 6,084 | |
| DTD [9] | | 47 | | | 1,880 | |
| Flowers102 [46] | Natural | 102 | 800/1000 | 200 | 6,149 | |
| Pets [49] | | 37 | | | 3,669 | |
| SVHN [45] | | 10 | | | 26,032 | |
| Sun397 [69] | | 397 | | | 21,750 | |
| Patch Camelyon [57] | | 2 | | | 32,768 | |
| EuroSAT [23] | Specialized | 10 | 800/1000 | 200 | 5,400 | Normalize by ImageNet Statistics |
| Resisc45 [8] | | 45 | | | 6,300 | |
| Retinopathy [29] | | 5 | | | 42,670 | |
| Clevr/count [28] | | 8 | | | 15,000 | |
| Clevr/distance [28] | | 6 | | | 15,000 | |
| DMLab [2] | | 6 | | | 22,735 | |
| KITTI/distance [16] | Structured | 4 | 800/1000 | 200 | 711 | |
| dSprites/location [43] | | 16 | | | 73,728 | |
| dSprites/orientation [43] | | 16 | | | 73,728 | |
| SmallNORB/azimuth [34] | | 18 | | | 12,150 | |
| SmallNORB/elevation [34] | | 9 | | | 12,150 | |

Table S3: Specifications of the few-shot datasets from FGVC. $\star$: Due to class imbalance, the training set size falls short of $k \times$ Class Number

| Dataset | Description | # Classes | Train size by $k$-shot | | | | Val | Test | Augmentation |
|---|---|---|---|---|---|---|---|---|---|
| | | | $k=1$ | $k=2$ | $k=4$ | $k=8$ | | | |
| CUB-200-2011 [59] | Bird species recognition | 200 | 200 | 400 | 800 | 1,600 | 600 | 5,794 | Resize to $256 \times 256$ |
| NABirds [56] | Bird species recognition | 555 | 555 | 1,110 | 2,220 | $4,427^\star$ | 2,393 | 24,633 | $\rightarrow$ Random crop to $224 \times 224$ |
| Oxford Flowers [46] | Flower species recognition | 102 | 102 | 204 | 408 | 816 | 1,020 | 6,149 | $\rightarrow$ Horizontal flip |
| Stanford Dogs [30] | Dog breed recognition | 120 | 120 | 240 | 480 | 960 | 1,200 | 8,580 | $\rightarrow$ Normalize (ImageNet stats) |
| Stanford Cars [15] | Car model classification | 196 | 196 | 392 | 784 | 1,568 | 815 | 8,041 | |

# E    Justification on SPT/rand

Although VIPAMIN outperforms the $K$-Means-based SPT results reported in the original paper on VTAB-1k, reproducing this initialization is computationally prohibitive for the full set of experiments we conduct. As an alternative, we adopt SPT/rand, which achieves the best performance among SPT variants according to Table 4(b) in [66]. However, as the reported results for SPT/rand are limited to only two datasets, we compare our reproduced SPT/rand results with the original SPT accuracy on VTAB-1k. The result can be found in Table S5.

We find that, with the MoCo-v3 backbone, SPT/rand performs comparably to the original SPT. In contrast, under the MAE-pretrained backbone, SPT/rand underperforms by approximately 2% in mean accuracy. To ensure consistency with the reported results and to maintain fidelity in baseline comparisons, we conduct our primary experiments using the MoCo-v3 pretrained model.

# F    Full Results on VTAB-1k

Table S6 presents the top-1 accuracy for full fine-tuning and VIPAMIN across the 19 individual datasets in VTAB-1k. For VIPAMIN, we report both the mean and standard deviation of top-1 accuracy, computed over three independent runs with different random seeds. When evaluated on MoCo-v3 pretrained models, VIPAMIN (shallow or deep) outperforms full fine-tuning in 16 out of 19 tasks. For MAE-pretrained models, it outperforms full fine-tuning in 14 out of 19 tasks. While the deep variant generally exhibits stronger performance than the shallow variant–likely due to increased representational capacity from higher prompt dimensionality–there are a few exceptions (e.g., Clevr/count on MAE, DMLab on MoCo-v3). Overall, these results demonstrate that VIPAMIN remains effective even when extended to deep prompt configurations.

Table S4: Hyperparameter tuning range

| | SPT/rand | VIPAMIN |
|---|---|---|
| Batch size | | 32 |
| Learning rate scheduler | | Cosine Decay with Linear Warmup |
| Total epochs | | 100 |
| Prompt length | | 100 |
| Optimizer | | AdamW |
| Learning rate | | $\{0.05, 0.1, 0.25, 0.5, 1.0, 2.5, 5.0\}$ |
| Weight decay | | 0.01 |
| $k$ | – | $\{2, 8, 32, 128\}$ |
| $\lambda$ | – | $\{0.0, 0.5, 1.0\}$ |

Table S5: Per-task fine-tuning for VTAB-1k benchmarks with pretrained ViT-B/16 as backbone. Bold indicates the better method between SPT and SPT/rand for each task.

| Methods | Natural (7) | | | | | | | Specialized (4) | | | | Structured (8) | | | | | | | | Mean |
|---|---|---|---|---|---|---|---|---|---|---|---|---|---|---|---|---|---|---|---|---|
| | Caltech101 | CIFAR-100 | DTD | Flowers102 | Pets | SVHN | Sun397 | Patch Camelyon | EuroSAT | Resisc45 | Retinopathy | Clevr/count | Clevr/distance | DMLab | KITTI/distance | dSprites/loc | dSprites/ori | SmallNORB/azi | SmallNORB/ele | |
| | | | | | | | | | | | | | | | | | | | | |
| *ViT-B with MoCo-v3 pretrained on ImageNet-1K* | | | | | | | | | | | | | | | | | | | | |
| SPT/rand | 89.8 | **59.3** | 69.2 | **92.7** | 87.9 | **83.7** | 39.7 | 83.2 | 94.6 | **84.5** | 71.3 | 56.4 | 59.5 | 45.0 | **79.9** | **75.7** | 48.4 | **29.7** | **46.5** | 68.3 |
| SPT | **91.0** | 58.1 | **69.6** | 91.1 | **89.4** | 82.2 | **39.9** | **83.6** | **94.7** | 82.0 | **75.4** | **73.3** | **60.6** | **45.7** | 71.4 | 75.0 | **42.1** | 28.0 | 45.2 | **68.3** |
| *ViT-B with MAE pretrained on ImageNet-1K* | | | | | | | | | | | | | | | | | | | | |
| SPT/rand | 83.8 | 25.2 | 57.7 | 72.2 | 71.2 | 76.0 | 20.8 | 77.4 | 91.9 | **73.0** | 73.3 | **73.6** | **61.7** | 40.5 | **75.0** | **72.8** | 44.6 | **27.9** | 33.0 | 60.6 |
| SPT | **87.6** | **29.5** | **61.5** | **77.4** | **80.8** | **77.2** | **23.7** | **84.4** | **93.0** | 70.8 | **75.4** | 73.3 | 55.5 | **44.0** | 73.2 | 70.6 | **48.0** | 27.4 | **35.7** | **62.6** |

## G  Full Results on FGVC Few-Shot Experiment

We report the full top-1 accuracy results, including standard deviations over three independent runs with different random seeds, for our few-shot experiments on the FGVC benchmark. This includes the shallow and deep variants of VPT, SPT/rand, and VIPAMIN across both MoCo-v3 and MAE pretrained models. Results for the MoCo-v3 pretrained models are summarized in Table S7. Among the 20 configurations evaluated (five datasets across four shot counts), VIPAMIN-Deep achieves the highest accuracy in 14 cases. Moreover, when averaged across the five datasets, VIPAMIN-Deep consistently outperforms all baselines at every shot level, underscoring its robustness and adaptability in data-scarce regimes.

A comparable trend is observed for MAE-pretrained models, as reported in Table S8. Here, VIPAMIN-Shallow obtains the best performance in 5 cases, while VIPAMIN-Deep achieves the highest accuracy in 11 cases, totaling 16 out of 20. In terms of mean accuracy across datasets, VIPAMIN-Deep again demonstrates superior performance at all shot levels.

These results collectively reinforce the effectiveness of VIPAMIN in alleviating the pronounced performance degradation typically observed in prompt tuning on self-supervised backbones under few-shot learning conditions. Its deep variant, in particular, exhibits consistent and substantial improvements over existing baselines, highlighting its capability to generalize across both pretraining paradigms and data scarcity scenarios.

## H  Adaptability of VIPAMIN

To examine how well our method works beyond pretrain-then-finetune image classification, we conducted multiple experiments on VIPAMIN under challenging conditions. These experiments included variations in model sizes, pretraining strategies, out-of-distribution generalization, and modality.

Table S6: Per-dataset performance comparison between Full fine-tuning result, VIPAMIN-Shallow, and VIPAMIN-Deep on VTAB-1k with MoCo-v3 ViT-B/16 and MAE ViT-B/16, over three independent runs. "†" indicates results reported in [66].

| Dataset | MoCo-v3 ViT-B/16 | | | MAE ViT-B/16 | | |
|---|---|---|---|---|---|---|
| | Full[†] | VIPAMIN Shallow | VIPAMIN Deep | Full[†] | VIPAMIN Shallow | VIPAMIN Deep |
| *Natural* | | | | | | |
| Caltech101 | 91.0 | **92.21** $\pm0.24$ | 91.05 $\pm0.40$ | 84.2 | 88.86 $\pm0.22$ | **89.01** $\pm0.52$ |
| CIFAR-100 | 57.6 | 71.34 $\pm0.30$ | **73.01** $\pm0.15$ | 24.6 | **35.95** $\pm0.52$ | 24.82 $\pm14.28$ |
| DTD | 64.6 | **69.96** $\pm0.28$ | 68.85 $\pm0.50$ | 56.9 | 58.92 $\pm0.44$ | **59.95** $\pm0.50$ |
| Flowers102 | 91.6 | 92.40 $\pm0.29$ | **93.95** $\pm0.09$ | 72.7 | 74.46 $\pm1.42$ | **75.53** $\pm0.43$ |
| Pets | 79.9 | 87.79 $\pm0.13$ | **87.97** $\pm0.04$ | 74.4 | 71.88 $\pm1.03$ | **79.53** $\pm0.10$ |
| SVHN | **89.8** | 86.53 $\pm0.06$ | 86.83 $\pm0.24$ | **86.6** | 80.04 $\pm1.09$ | 81.83 $\pm1.80$ |
| Sun397 | 29.1 | 37.04 $\pm0.12$ | **42.13** $\pm0.02$ | 15.8 | 23.05 $\pm0.12$ | **23.17** $\pm0.40$ |
| *Specialized* | | | | | | |
| Patch Camelyon | 85.1 | 82.83 $\pm0.26$ | **85.89** $\pm0.54$ | 81.8 | 80.13 $\pm0.94$ | **84.24** $\pm0.46$ |
| EuroSAT | 96.4 | 96.26 $\pm0.05$ | **96.61** $\pm0.14$ | **94.0** | 92.78 $\pm0.53$ | 93.52 $\pm2.52$ |
| Resisc45 | 83.1 | 83.07 $\pm0.24$ | **84.80** $\pm0.16$ | 72.3 | 73.26 $\pm0.12$ | **75.23** $\pm0.33$ |
| Retinopathy | 74.2 | **74.39** $\pm0.20$ | 71.84 $\pm0.73$ | 70.6 | **73.66** $\pm0.14$ | 68.66 $\pm1.84$ |
| *Structured* | | | | | | |
| Clevr/count | 55.2 | 69.23 $\pm0.72$ | **74.65** $\pm0.95$ | 67.0 | **73.65** $\pm0.42$ | 70.61 $\pm3.24$ |
| Clevr/distance | 56.9 | 59.28 $\pm0.62$ | **63.54** $\pm0.70$ | **59.8** | 57.62 $\pm0.68$ | 56.82 $\pm3.56$ |
| DMLab | 44.6 | **47.56** $\pm0.14$ | 47.10 $\pm0.55$ | **45.2** | 43.37 $\pm0.29$ | 44.42 $\pm3.18$ |
| KITTI/distance | 77.9 | 78.25 $\pm1.52$ | **79.05** $\pm0.60$ | 75.3 | 78.06 $\pm1.47$ | **80.03** $\pm1.22$ |
| dSprites/loc | 63.8 | 75.82 $\pm0.84$ | **84.49** $\pm2.41$ | 72.5 | 80.02 $\pm1.96$ | **82.26** $\pm1.32$ |
| dSprites/ori | **49.0** | 47.81 $\pm1.61$ | 47.15 $\pm0.44$ | 47.5 | 46.96 $\pm0.57$ | **49.50** $\pm0.35$ |
| SmallNORB/azi | **31.5** | 26.74 $\pm0.31$ | 29.27 $\pm0.35$ | **30.2** | 26.92 $\pm0.66$ | 23.23 $\pm0.20$ |
| SmallNORB/ele | 36.9 | **48.74** $\pm2.08$ | 45.17 $\pm0.47$ | 33.0 | **53.13** $\pm1.13$ | 46.39 $\pm0.97$ |

## H.1 Robustness under Distribution Shift

To test whether our approach is confined to the standard pretrain-then-finetune paradigm, we evaluate Out-of-Distribution (OOD) generalization, a stringent check of robustness beyond in-distribution tuning. Specifically, we corrupt the CIFAR-100 test set with multiple distribution shifts and compare against baselines. As summarized in Table S9, our method attains the best accuracy on 5 of 6 corruptions and the highest mean accuracy, improving by 4.0% over SPT/rand. These results indicate that the benefits of our method extend beyond the pretrain-then-finetune setting, translating into stronger performance under distribution shift.

## H.2 Generalization to Self-Supervised Models and Scales

While our main paper primarily evaluates VIPAMIN using MAE and MoCo-v3 with ViT-Base and larger models, we further investigate its adaptability across different self-supervised pretraining paradigms and model scales. To this end, we experiment with DINO ViT-Small/16 and MAE ViT-Tiny/16, representing the regimes of contrastive learning and masked image modeling, respectively [5, 64].

DINO distinguishes itself from MoCo-v3 by employing self-distillation via a momentum-updated teacher, while still retaining a contrastive training structure. Notably, DINO achieves 81.5% Top-1 accuracy on ImageNet with ViT-S/16 under fine-tuning. On the other hand, MAE-Tiny, although not included in the original MAE paper [22], has been adapted in [64], which introduces a tailored pretraining strategy for smaller architectures. This setup yields 78.0% Top-1 accuracy on ImageNet with ViT-Tiny.

Table S10 presents the results on VTAB-1k. For DINO ViT-Small/16, VIPAMIN outperforms full fine-tuning for 16 out of 19 tasks, highlighting its strong adaptability to smaller architectures. However, in

Table S7: MoCo-v3 few-shot accuracy (%) for $k$-shot classification across FGVC datasets over three independent runs.

| $k$-shot | Method | Mode | CUB | NABirds | Flowers | Dogs | Cars | Mean |
|---|---|---|---|---|---|---|---|---|
| 1 | VPT | Shallow | 15.67±0.34 | 7.68±0.35 | 31.42±1.15 | 31.18±2.41 | 4.74±0.10 | 18.14 |
| | | Deep | 16.87±0.06 | 8.96±0.39 | 36.51±21.74 | 33.87±0.05 | 4.60±0.15 | 20.16 |
| | SPT/rand | Shallow | 17.16±0.31 | 11.67±0.42 | 48.90±0.79 | 35.52±0.31 | 5.29±0.02 | 23.71 |
| | | Deep | 21.37±0.06 | 13.46±0.20 | 53.07±0.28 | 36.41±0.25 | 5.70±0.07 | 26.00 |
| | VIPAMIN | Shallow | 20.14±0.12 | 12.61±0.08 | 52.81±0.05 | **37.47**±0.40 | 5.73±0.14 | 25.75 |
| | | Deep | **21.65**±0.13 | **13.65**±0.08 | **53.66**±0.32 | 36.89±0.27 | **5.92**±0.13 | **26.35** |
| 2 | VPT | Shallow | 15.59±10.07 | 11.69±0.21 | 59.03±0.94 | 45.40±1.73 | 6.40±0.12 | 27.62 |
| | | Deep | 38.19±1.67 | 7.40±9.06 | 72.22±0.30 | 34.50±23.56 | 9.20±0.38 | 32.30 |
| | SPT/rand | Shallow | 29.83±0.32 | 22.54±0.48 | 70.44±0.22 | 48.97±0.19 | 10.90±0.05 | 36.54 |
| | | Deep | **38.65**±0.16 | 26.13±0.10 | 75.00±0.29 | 50.15±0.00 | 11.78±0.09 | 40.34 |
| | VIPAMIN | Shallow | 35.97±0.43 | 23.14±0.28 | 71.62±0.21 | 49.39±0.12 | 11.06±0.07 | 38.24 |
| | | Deep | 38.53±0.19 | **26.92**±0.09 | **75.42**±0.09 | **50.86**±0.19 | **12.05**±0.11 | **40.76** |
| 4 | VPT | Shallow | 31.35±0.65 | 14.29±0.04 | 66.20±0.10 | 36.81±25.02 | 9.88±0.14 | 31.71 |
| | | Deep | 57.71±0.17 | 29.28±0.56 | 83.69±1.09 | **63.06**±0.15 | 8.98±7.87 | 48.54 |
| | SPT/rand | Shallow | 51.73±0.74 | 40.46±0.13 | 84.64±0.07 | 59.84±0.26 | 21.66±0.15 | 51.67 |
| | | Deep | **58.10**±0.22 | 44.81±0.18 | 84.48±1.33 | 60.74±0.12 | 24.12±0.07 | 54.45 |
| | VIPAMIN | Shallow | 53.65±0.74 | 40.96±0.09 | 85.19±0.14 | 60.25±0.15 | 21.88±0.11 | 52.39 |
| | | Deep | 58.01±0.12 | **44.92**±0.05 | **85.79**±1.27 | 61.27±0.17 | **25.55**±0.05 | **55.11** |
| 8 | VPT | Shallow | 37.25±0.22 | 17.22±0.14 | 77.79±0.39 | 62.79±0.23 | 13.46±0.40 | 41.70 |
| | | Deep | 62.78±0.75 | 31.24±0.15 | 89.48±0.01 | 55.95±15.50 | 26.33±15.97 | 53.16 |
| | SPT/rand | Shallow | 66.58±0.27 | 54.96±0.21 | 92.90±0.04 | 69.14±0.07 | 43.80±0.32 | 65.48 |
| | | Deep | **70.11**±0.13 | 58.40±0.11 | 94.92±0.21 | **70.25**±0.08 | 48.64±0.32 | 68.46 |
| | VIPAMIN | Shallow | 68.55±0.22 | 55.12±0.06 | 94.34±0.08 | 70.03±0.00 | 43.83±0.13 | 66.37 |
| | | Deep | 69.63±0.14 | **58.68**±0.09 | **95.51**±0.15 | 69.66±0.24 | **51.01**±0.28 | **68.90** |

Table S8: MAE Few-shot accuracy (%) for $k$-shot classification across FGVC datasets over three independent runs.

| $k$-shot | Method | Mode | CUB | NABirds | Flowers | Dogs | Cars | Mean |
|---|---|---|---|---|---|---|---|---|
| 1 | VPT | Shallow | 2.28±0.06 | 0.80±0.02 | 13.26±1.51 | 1.08±0.13 | 0.96±0.13 | 3.68 |
| | | Deep | 2.03±0.08 | 0.78±0.01 | 12.59±1.10 | 2.55±0.22 | 1.61±0.09 | 3.91 |
| | SPT/rand | Shallow | 3.65±0.04 | 1.67±0.04 | 30.04±0.29 | 7.27±0.24 | 2.90±0.08 | 9.91 |
| | | Deep | 4.07±0.09 | 2.15±0.10 | 29.10±0.72 | 10.21±0.09 | 2.79±0.09 | 9.66 |
| | VIPAMIN | Shallow | 4.16±0.21 | 1.90±0.18 | **30.56**±0.15 | 8.29±0.03 | **2.95**±0.09 | 9.97 |
| | | Deep | **4.55**±0.08 | **2.21**±0.07 | 30.35±0.45 | **11.32**±0.26 | 2.83±0.10 | **10.65** |
| 2 | VPT | Shallow | 3.80±0.28 | 2.26±0.23 | 28.16±4.51 | 8.40±1.77 | 3.30±0.22 | 9.58 |
| | | Deep | 3.64±0.06 | 2.40±0.08 | 32.17±0.51 | 6.21±0.75 | 3.62±0.05 | 9.61 |
| | SPT/rand | Shallow | 6.19±0.25 | 4.40±0.11 | 46.02±0.91 | 16.43±0.32 | 5.54±0.13 | 15.72 |
| | | Deep | 7.38±0.30 | 4.70±0.07 | 46.38±0.41 | 23.28±0.29 | 5.21±0.37 | 17.39 |
| | VIPAMIN | Shallow | 7.85±0.49 | **4.84**±0.34 | 46.53±1.65 | 17.40±0.21 | 5.21±0.25 | 16.37 |
| | | Deep | **8.30**±0.16 | 4.61±0.12 | **48.43**±0.50 | **23.57**±0.34 | **5.84**±0.10 | **18.15** |
| 4 | VPT | Shallow | 9.12±0.91 | 5.22±0.16 | 46.70±0.44 | 23.28±0.46 | 4.84±0.46 | 17.43 |
| | | Deep | 12.77±0.52 | 11.13±0.23 | 54.19±1.56 | 18.30±4.88 | 8.12±0.48 | 20.10 |
| | SPT/rand | Shallow | 17.32±0.51 | 13.19±0.29 | 67.42±0.13 | 31.21±0.37 | 8.74±0.18 | 27.18 |
| | | Deep | **18.91**±0.48 | 14.38±0.31 | 66.99±0.44 | **39.45**±0.23 | 9.37±0.37 | 29.82 |
| | VIPAMIN | Shallow | 18.82±0.32 | **14.49**±0.29 | 67.32±0.91 | 35.29±0.68 | 8.77±0.04 | 28.14 |
| | | Deep | 17.99±0.28 | 14.28±0.32 | **69.20**±0.12 | 38.93±0.44 | **9.78**±0.18 | **30.04** |
| 8 | VPT | Shallow | 14.53±1.35 | 10.70±0.73 | 63.03±0.91 | 28.68±15.84 | 8.31±0.40 | 25.45 |
| | | Deep | 18.80±3.90 | 22.71±12.45 | 76.63±0.89 | 16.00±5.20 | 20.31±0.30 | 30.49 |
| | SPT/rand | Shallow | 37.31±0.70 | 31.30±0.20 | 80.72±0.48 | 49.48±0.12 | 25.42±0.89 | 44.05 |
| | | Deep | **40.50**±0.12 | **36.12**±0.43 | 82.61±0.59 | 53.66±0.11 | 25.60±0.34 | 47.30 |
| | VIPAMIN | Shallow | 37.55±0.78 | 31.77±0.38 | 82.05±0.57 | 53.21±0.55 | **28.54**±0.39 | 46.22 |
| | | Deep | 39.22±0.81 | 34.22±0.36 | **84.11**±0.21 | **53.98**±0.38 | 27.13±0.09 | **47.33** |

the case of the smallest model, MAE ViT-Tiny/16, VIPAMIN falls short of full fine-tuning. While VIPAMIN still achieves substantial improvements over VPT on MAE ViT-Tiny/16 (as shown later in Figure S1), its relative underperformance in this low-capacity regime remains an open question.

We hypothesize that the limited embedding dimensionality–for instance, only 16 dimensions per head in the 12-head ViT-Tiny/16 model–restricts representational flexibility. In such settings, the injection of prompt tokens may dilute critical interactions among patch embeddings, ultimately leading to suboptimal performance.

Table S9: OOD Generalization test on CIFAR-100. Used ViT-B model pretrained by MoCo-v3.

| | Brightness | Contrast | Rotation | Gaussian | Shot | Speckle | Mean |
|---|---|---|---|---|---|---|---|
| VPT | 36.2 | 20.9 | 6.80 | 13.2 | 21.4 | 25.5 | 20.6 |
| SPT/rand | 65.5 | 47.2 | 16.8 | **20.2** | 37.0 | 44.5 | 38.5 |
| VIPAMIN | **71.4** | **57.5** | **18.7** | 19.8 | **39.9** | **48.8** | **42.5** |

Furthermore, we investigate the extension of VIPAMIN to hierarchical transformers–vision backbones that build multi-scale features via staged downsampling (a pyramid of token resolutions) rather than a single, fixed-resolution sequence. This structure enhances efficiency and is standard in tasks such as object detection [44]. We instantiate this with HiViT [74], selected for its explicit pairing of hierarchical transformers with masked-image modeling (MIM) pretraining. Unlike typical hierarchical ViTs, HiViT drops the windowed attention and patch merging stage. As shown in Table S11, VIPAMIN outperforms baselines on 12 of 19 tasks, demonstrating effective transfer to hierarchical backbones and highlighting its adaptability.

Table S10: Per-dataset performance comparison between Full fine-tuning and VIPAMIN(-Shallow) on VTAB-1k with DINO ViT-Small/16 and MAE ViT-Tiny/16, over three independent runs. For full fine-tuning, we search learning rate from $[0.0001, 0.0005, 0.001, 0.005]$ following [27].

| Dataset | DINO ViT-Small/16 | | MAE ViT-Tiny/16 | |
|---|---|---|---|---|
| | Full | VIPAMIN | Full | VIPAMIN |
| *Natural* | | | | |
| Caltech101 | 73.41 ±1.32 | **83.01** ±1.50 | **61.42** ±1.37 | 51.69 ±0.33 |
| CIFAR-100 | 26.32 ±2.88 | **51.13** ±0.93 | **17.51** ±1.26 | 16.09 ±0.10 |
| DTD | 54.24 ±0.82 | **60.44** ±0.55 | **38.58** ±0.63 | 31.22 ±0.23 |
| Flowers102 | 80.20 ±1.68 | **84.28** ±0.70 | **55.78** ±1.69 | 46.60 ±0.82 |
| Pets | 72.37 ±1.54 | **82.24** ±0.57 | **29.82** ±1.09 | 21.19 ±0.59 |
| SVHN | **84.32** ±0.68 | 58.85 ±27.32 | **84.29** ±0.57 | 69.30 ±1.21 |
| Sun397 | 14.98 ±1.02 | **28.81** ±0.40 | 7.56 ±0.36 | **8.73** ±0.23 |
| *Specialized* | | | | |
| Patch Camelyon | **84.04** ±2.47 | 78.06 ±1.97 | **78.69** ±1.09 | 75.51 ±0.83 |
| EuroSAT | **95.61** ±0.51 | 92.61 ±0.44 | **92.17** ±0.20 | 88.57 ±0.38 |
| Resisc45 | **77.30** ±0.70 | 75.44 ±0.62 | **61.96** ±0.26 | 53.39 ±0.48 |
| Retinopathy | 72.65 ±0.75 | **73.36** ±0.24 | 70.98 ±0.65 | **71.81** ±0.42 |
| *Structured* | | | | |
| Clevr/count | 43.31 ±0.99 | **70.73** ±0.50 | **62.67** ±1.68 | 62.21 ±0.43 |
| Clevr/distance | 49.42 ±0.70 | **52.70** ±0.80 | **61.51** ±0.46 | 60.40 ±1.29 |
| DMLab | 43.30 ±0.52 | **44.49** ±0.57 | **40.96** ±0.78 | 37.19 ±0.80 |
| KITTI/distance | **77.68** ±0.86 | 70.37 ±0.57 | **71.82** ±2.12 | 65.07 ±2.20 |
| dSprites/loc | 46.48 ±1.24 | **73.85** ±0.45 | **78.88** ±0.36 | 76.11 ±0.79 |
| dSprites/ori | 37.24 ±0.28 | **40.92** ±2.24 | **37.03** ±0.37 | 34.17 ±0.95 |
| SmallNORB/azi | **24.82** ±1.63 | 20.98 ±0.09 | **27.44** ±0.84 | 23.25 ±0.57 |
| SmallNORB/ele | 29.60 ±0.92 | **34.42** ±0.65 | 40.21 ±1.07 | **45.21** ±0.52 |

## H.3 Transferability to Language Tasks

While our motivation in the main paper is limited to visual tasks, we also test our method in the language domain. We apply our method to BERT-Large [11] on the SuperGLUE benchmark [61] to test generality across modality and scale. SuperGLUE is an extensive benchmark including the following tasks–natural language inference (RTE, CB), coreference resolution (WSC), sentence completion (COPA), word sense disambiguation (WiC), and question answering (MultiRC, ReCoRD, BoolQ). As shown in Table S12, our method outperforms both P-Tuning v2 [40] and full fine-tuning, two of the strongest baselines in the language domain. This is an intriguing result, as it suggests that our motivation regarding selective token attention and the injection of new knowledge can naturally extend to the language domain as well.

Table S11: Per-dataset performance comparison between baselines and VIPAMIN(-Shallow) on VTAB-1k with HiViT, over three independent runs.

| Dataset | VPT | SPT/rand | VIPAMIN |
|---|---|---|---|
| *Natural* | | | |
| Caltech101 | 88.26 ±0.33 | 88.33 ±0.20 | **88.35** ±0.38 |
| CIFAR-100 | 40.41 ±0.96 | 40.45 ±1.34 | **41.05** ±0.76 |
| DTD | 59.98 ±0.18 | 60.00 ±0.22 | **60.50** ±0.72 |
| Flowers102 | 74.65 ±0.78 | **76.44** ±0.08 | 76.33 ±0.68 |
| Pets | 76.71 ±0.22 | **76.91** ±0.41 | 76.47 ±0.07 |
| SVHN | 75.61 ±1.13 | 81.58 ±0.61 | **82.60** ±0.13 |
| Sun397 | 22.18 ±0.43 | 22.14 ±0.15 | **22.33** ±0.30 |
| *Specialized* | | | |
| Patch Camelyon | 78.44 ±0.36 | 78.40 ±0.36 | **79.52** ±1.10 |
| EuroSAT | 93.36 ±0.12 | 93.84 ±0.20 | **94.65** ±0.30 |
| Resisc45 | 68.96 ±0.72 | **70.40** ±0.57 | 70.22 ±0.77 |
| Retinopathy | 73.80 ±0.05 | **74.44** ±0.21 | 74.28 ±0.37 |
| *Structured* | | | |
| Clevr/count | 67.26 ±0.27 | **69.46** ±0.24 | 69.14 ±0.46 |
| Clevr/distance | **62.59** ±0.34 | 61.65 ±0.36 | 62.23 ±0.44 |
| DMLab | 45.12 ±0.27 | 45.36 ±0.50 | **46.16** ±0.31 |
| KITTI/distance | 79.93 ±0.93 | 81.62 ±0.46 | **82.23** ±0.29 |
| dSprites/loc | 83.72 ±0.72 | 82.94 ±0.52 | **86.49** ±0.46 |
| dSprites/ori | 25.79 ±2.78 | 29.32 ±1.88 | **51.44** ±0.88 |
| SmallNORB/azi | 21.50 ±1.21 | **24.21** ±0.30 | 22.61 ±0.18 |
| SmallNORB/ele | 39.53 ±0.48 | 41.20 ±0.32 | **44.02** ±1.42 |

Table S12: Per-task results for SuperGLUE development set with a pretrained BERT-Large. "†" indicates results reported in [72].

| BERT-L (335M) | BoolQ | CB | COPA | MultiRC | ReCoRD | RTE | WiC | WSC | Mean |
|---|---|---|---|---|---|---|---|---|---|
| Full[†] | 77.7 | 94.6 | 69.0 | 70.5 | 70.6 | 70.4 | 74.9 | 68.3 | 74.5 |
| Prompt Tuning[†] | 67.2 | 80.4 | 55.0 | 59.6 | 44.2 | 53.5 | 63.0 | 64.4 | 60.9 |
| P-Tuning v2[†] | 73.1 | 94.6 | 73.0 | **70.6** | 72.8 | 78.3 | 75.1 | 68.3 | 75.7 |
| E$^2$VPT[†] | 74.4 | 80.4 | 77.0 | 65.8 | 71.9 | 78.7 | 74.3 | 67.3 | 73.7 |
| VFPT[†] | **74.8** | 81.2 | 78.1 | 67.8 | **72.9** | 77.2 | **75.3** | 68.4 | 74.5 |
| VIPAMIN-Deep | 74.6 | **94.6** | **79.0** | 66.4 | 70.6 | **79.1** | 74.3 | **69.2** | **76.0** |

# I Further Analyses

## I.1 Extensive Comparison with VPT

Continuing from Figure 1(b) of our main paper, we report the impact of VIPAMIN in terms of LP ratio (i.e., a proxy for task similarity to the pretraining task), and how it effectively counteracts the underperformance of vanilla VPT.

Figure S1 presents the results. Two key observations emerge. First, the underperformance of VPT on tasks with low LP ratios is a consistent trend across different self-supervised pretraining strategies and model scales. Second, while VIPAMIN consistently improves upon VPT across all LP ratio regimes, its gains are particularly pronounced on tasks with lower LP ratios. This suggests that addressing uniform attention patterns and prompt subspace collapse is crucial for enhancing the adaptability of VPT, especially in tasks that exhibit significant distributional shifts from the pretraining domain.

## I.2 GradCAM Analysis

For interpretability, we visualize the GradCAM maps of the trained models on various datasets. Results are shown in Figure S2. Two distinct failure modes of VPT become apparent, each manifesting differently depending on the task similarity to the pretraining distribution. For tasks that are similar to the pretraining domain, VPT exhibits overly uniform attention across the image, failing to localize semantically meaningful regions. In contrast, VIPAMIN produces sharper and more localized attention maps, owing to its matching module, which encourages prompt specialization through semantically coherent token alignment.

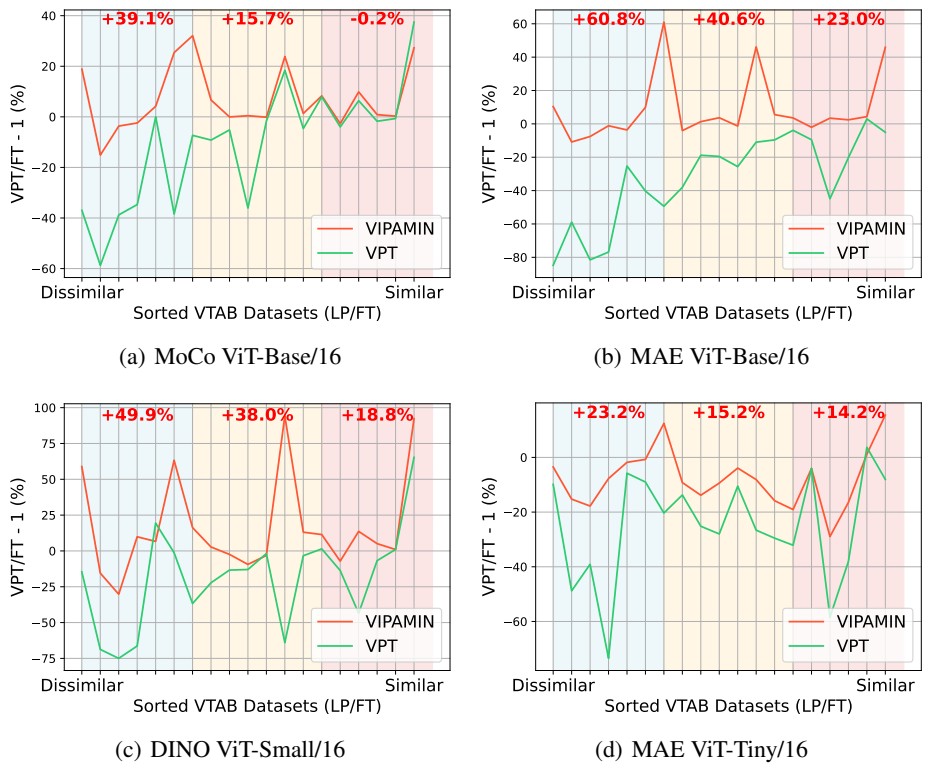

(a) MoCo ViT-Base/16      (b) MAE ViT-Base/16

(c) DINO ViT-Small/16      (d) MAE ViT-Tiny/16

Figure S1: Relative accuracy ratio of VPT and VIPAMIN compared to full fine-tuning across 19 VTAB datasets, sorted by the LP ratio. The three percentages highlighted in red at the top of the figure denote the mean improvement of VIPAMIN over VPT within each of the three LP ratio segments–low, medium, and high–as defined by the tertile split of the dataset ordering.

On dissimilar tasks, VPT often fails to grasp the task semantics, resulting in attention focused on irrelevant or background regions. This suggests an inability to adaptively reconfigure the prompt representation. VIPAMIN overcomes this limitation by explicitly injecting novel representational directions into the prompt space via its orthogonalizing module. As a result, VIPAMIN consistently attends to task-relevant regions, demonstrating improved interpretability and alignment with human intuition.

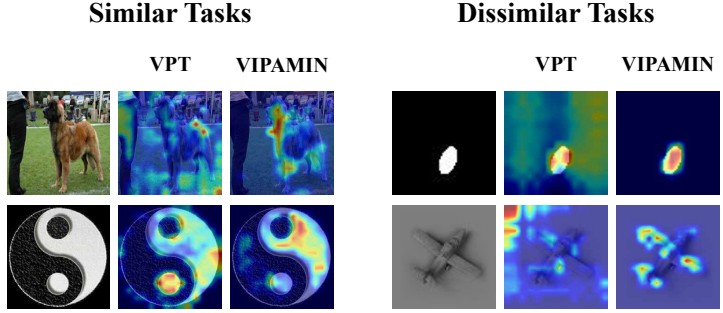

Figure S2: GradCAM visualization result of DINO ViT-Small/16 on various VTAB-1k datasets.

## I.3   Feature Selection in Matching Module

The matching module in VIPAMIN relies on key-space refined features ($\mathbf{Z}_0 \mathbf{W}_K$) to compute similarity between prompts and input tokens. To assess the impact of the feature type used for matching, we

conduct an ablation study comparing various reference features extracted from the first transformer block:

$$\mathrm{SA}(\mathbf{Z}_0) = \mathrm{softmax}\left(\frac{\mathbf{Z}_0\mathbf{W}_Q(\mathbf{Z}_0\mathbf{W}_K)^T}{\sqrt{d}}\right)\mathbf{Z}_0\mathbf{W}_V$$

$$\mathbf{Z}_1 = B_1(\mathbf{Z}_0)$$

When using the attention weights (blue box) as a reference, we directly apply the softmax scores for patch selection. For other feature options (black boxes), we compute cosine similarity to identify the top-$k$ tokens. As shown in Table S13, using $\mathbf{Z}_0\mathbf{W}_K$ yields the best performance, while attention-based selection performs the worst. This suggests that explicitly computing similarity in a shared projected space is critical to the effectiveness of VIPAMIN.

Interestingly, this observation aligns with findings from token merging studies in vision transformers [3], which aim to accelerate inference by merging redundant tokens. In that context, key vectors ($\mathbf{E}_0\mathbf{W}_K$) have been shown to effectively summarize token content for measuring redundancy. Similarly, our use of the key space exploits the natural clustering of semantically related tokens, enabling more coherent and task-relevant prompt initialization.

Table S13: **Ablation on feature selection.** MoCo-v3 pretrained model is trained and evaluated on *Specialized* tasks in VTAB-1k. Hyperparameter $\lambda$ is set to zero for clarity.

| Feature | Mean Acc |
|---|---|
| $\mathbf{Z}_0$ | 82.36 |
| $\mathbf{Z}_0\mathbf{W}_Q$ | 82.39 |
| $\mathbf{Z}_0\mathbf{W}_K$ | **82.85** |
| softmax | 81.80 |
| $\mathbf{Z}_0\mathbf{W}_V$ | 82.23 |
| $\mathbf{Z}_1$ | 82.22 |

## I.4 Role of Hyperparameters in VIPAMIN

Extending the analysis from Section 5.4, we investigate the functional roles of VIPAMIN's two core components by examining trends in optimal hyperparameter settings across tasks. VIPAMIN introduces two key hyperparameter: $k$, which specifies the number of tokens to which each prompt attends, and $\lambda$, which controls the degree of orthogonal bias injected into the prompt initialization.

Figure S3 (a) and (b) present results from VTAB-1k and FGVC benchmarks, respectively. For tasks that exhibit significant distributional shift from the pretraining domain, smaller values of $k$ and $\lambda$ approaching 1 yield the best performance. This suggests that such tasks benefit from more localized attention and stronger orthogonalization, which together enable the model to focus on task-specific features not captured by the pretrained backbone. In contrast, tasks more closely aligned with the pretraining distribution tend to favor larger $k$ and smaller $\lambda$, indicating that broader attention and greater reliance on pretrained representations are advantageous.

In few-shot scenarios, similar trends are observed for $\lambda$ as the number of shots increases, implying that orthogonal bias provides useful inductive structure when prompt parameters cannot be fully optimized due to limited supervision. While no definitive trend is observed for $k$ in the few-shot setting, larger values generally lead to robust performance across varying shot counts.

## I.5 Extension toward VPT-Deep

Empirically, we demonstrate that VIPAMIN yields performance gains even under the VPT-Deep variant as shown in Sec. 5.3, raising the question of whether prompt subspace collapse persists in deeper architectures. To check this, for each transformer block $l \in [L]$, with input $[\mathbf{P}_{l-1}; \mathbf{X}_{l-1}]$, we compute the projection energy, as defined in (6),

$$\mathrm{ProjectionEnergy}((\mathbf{P}_{l-1}\mathbf{W}_V^l)^T \rightarrow (\mathrm{SA}_l(\mathbf{X}_{l-1}))^T), \tag{11}$$

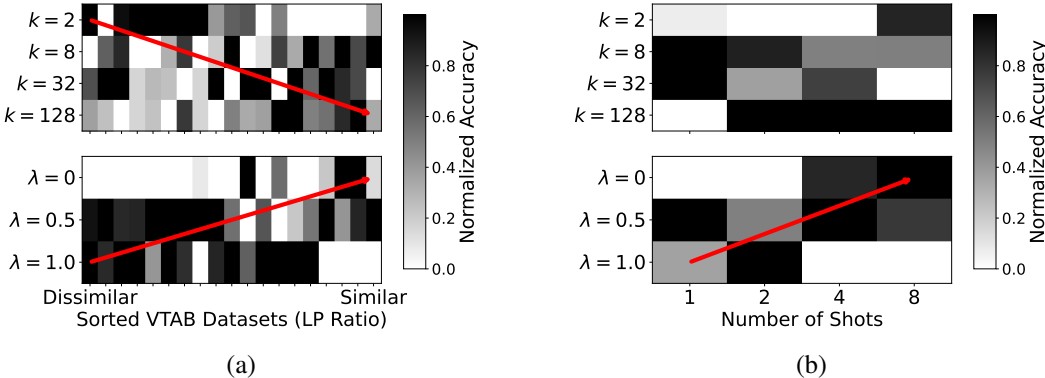

(a)            (b)

Figure S3: **Effect of Hyperparameter** (a) We evaluate the performance of MAE pretrained VIPAMIN on VTAB-1k, across varying values of $k, \lambda$. Datasets are sorted by their LP ratio (i.e., the relative closeness of linear probing to full fine-tuning performance) from lowest to highest. (b) We evaluate the few-shot performance of MoCo-v3 pretrained VIPAMIN on NABirds. The $x$-axis represents the number of shots. For both figures (a) and (b), we select the best learning rate based on peak validation accuracy, and normalize the resulting accuracy to $[0, 1]$ range per dataset (column). We present rough trend line of best performing hyperparameters in red arrow.

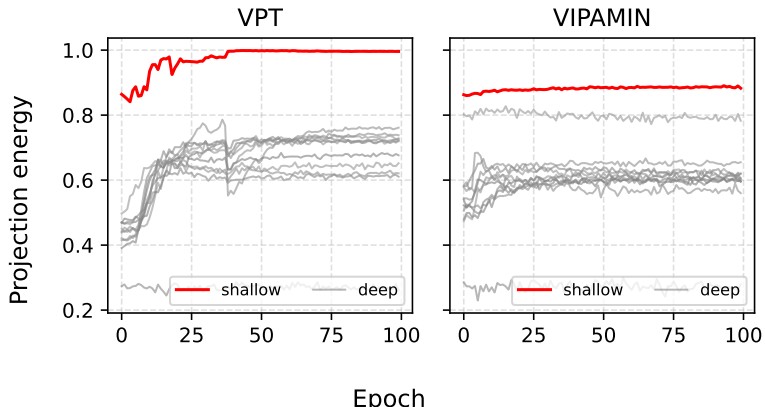

Figure S4: Deep projection energy trend during training dSprites/loc–the most dissimilar task.

where $\mathbf{W}_V^l$ denotes the value projection matrix in the $l$-th block and $\mathrm{SA}_l(\cdot)$ is the self-attention module of $l$-th block. As shown in Figure S4, prompt collapse is observed only in the first layer–where the initial shallow prompt is prepended–and does not occur in subsequent layers. This pattern suggests that the performance improvements of VIPAMIN-Deep may be largely attributed to preventing collapse at the initial block. However, we posit that deeper orthogonality still plays a crucial role. Through the lens of representation oversmoothing in deep transformers, prior work [12, 63] shows that self-attention layers reduce representational rank and act as low-pass filters; our orthogonalization module explicitly increases the rank of self-attention outputs, counteracting this depth-wise tendency. Thus, although Figure S4 illustrates only early-layer prompt-side collapse, deep orthogonalization likely also mitigates oversmoothing of the attention pathway–a complementary failure mode consistent with the prior theory.

## I.6 Supervised VPT vs. Self-Supervised VPT

Our study primarily investigates visual prompt initialization for VPT in self-supervised backbones. A natural extension is to ask whether similar failure modes–such as uniform prompt attention–also arise in supervised VPT. Figure S5 presents a comparison of prompt attention entropy across the 19 VTAB-1k tasks (ordered by LP Ratio), contrasting VPT with self-supervised (MoCo-v3) and

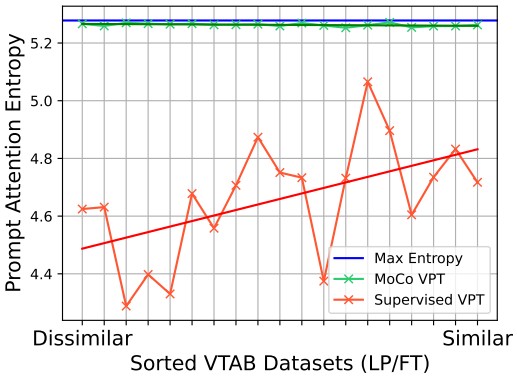

Figure S5: Comparison of prompt attention entropy (5) across VTAB-1k tasks for VPT using MoCo-v3 (self-supervised) and supervised pretrained backbones.

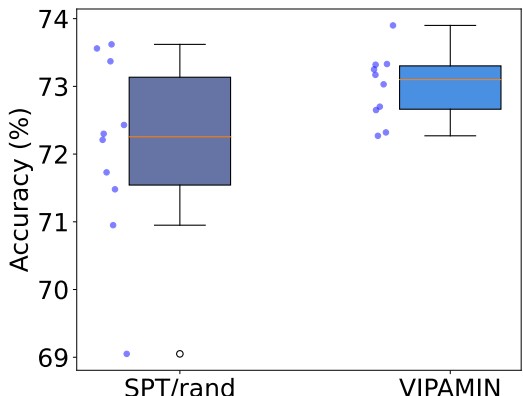

Figure S6: We conduct 10 independent training with the best performing hyperparameter and report the distribution of test accuracy of MAE pretrained model on Resisc45 dataset.

supervised backbones. The results show that VPT with a supervised backbone consistently exhibits lower entropy, indicating a greater tendency to focus attention on a smaller subset of tokens.

We attribute this behavior to the nature of semantic representations learned through supervised pretraining. Prior work has shown that supervised ViTs tend to encode more class-aligned and semantically structured features at the image level [60]. We hypothesize that, in this setting, even randomly initialized prompts can more easily attend to meaningful regions, thereby reducing the benefits of specialized prompt initialization techniques.

## I.7   Stability Study

Prompt tuning is known to exhibit instability across random seeds, raising concerns about its reproducibility [6]. One key source of variability is the order in which training data is presented during optimization. To evaluate robustness to this factor, we conduct 10 independent runs of each method with different data orderings while keeping all other settings fixed, including the initialization point. As shown in Figure S6, VIPAMIN not only achieves the highest average accuracy but also exhibits significantly lower variance, indicating improved stability across random seeds.

