# OpenReview forum: "VIPAMIN: Visual Prompt Initialization via Embedding Selection and Subspace Expansion"
_NeurIPS.cc/2025/Conference — NeurIPS 2025 poster_

### Official Review · Reviewer_MNuV · 2025-06-06

**Clarity:** 2
**Significance:** 2
**Originality:** 2
**Rating:** 4
**Confidence:** 2

**Summary:**

Identifying the uniform attention and subspace collapse limitation, this paper introduces VIPAMIN, a initialization technique that addresses these issues via attention-guided matching and orthogonal subspace injection. Specifically, VIPAMIN aligns prompts with semantically informative regions in the embedding space, and injects representational directions beyond the pretrained subspace. VIPAMIN are validated under few-shot settings.

**Questions:**

1. In Figure 1, the importance of “semantic token selection” and “bias injection,” are observed, but is there a more explicit visualization? For example, which specific semantic tokens are selected, and what precise biases are being injected?
2. In Figure 2, the “Failure modes of VPT” are mentioned. To which modes does “mode” refer? Please clarify the specific failure modes being illustrated.
3. In Table 3, what does the metric labeled “Spec” denote? Its performance is inferior to several baselines—can the authors provide a detailed analysis explaining this discrepancy?

**Ethical Concerns:**

["NO or VERY MINOR ethics concerns only"]

**Final Justification:**

Thank you! I keep my current rating.

**Limitations:**

yes.

**Paper Formatting Concerns:**

There are no apparent formatting issues that require special attention.

**Quality:**

3

**Strengths And Weaknesses:**

**Strengths**

1. The paper introduces several novel technical modules—specifically, “Selecting Tokens for Prompt Propagation” and “Injecting Semantic Bias into Attention Outputs”—both of which represent innovative contributions.

**Weaknesses**

1. **Lack of Detailed Visualization for Semantic Token Selection and Bias Injection**
   Although Figure 1 suggests the importance of “semantic token selection” and “bias injection,” the paper does not provide explicit visualizations that identify which tokens are being selected or which biases are being injected. Without such detailed illustrations, it is difficult to evaluate whether the selected tokens indeed capture the intended semantics or whether the injected biases align with the theoretical motivation.

2. **Unclear Definition of “Failure Modes” in VPT**
   The term “mode” in Figure 2’s “Failure modes of VPT” is not defined, leaving the reader uncertain about the specific failure cases being depicted. This ambiguity prevents a clear understanding of the limitations of VPT and hinders the ability to assess whether the proposed method effectively addresses those failure modes.

3. **Ambiguity Surrounding the “Spec” Metric in Table 3**
   Table 3 reports a metric labeled “Spec,” but the paper does not define this term. Furthermore, “Spec” underperforms relative to baseline methods, yet no detailed analysis is provided to explain why. The absence of a clear definition and an in-depth investigation into the poorer performance undermines the reader’s ability to interpret the results and understand the method’s limitations.

---

> ### Author Rebuttal · Authors · 2025-07-31
>
> We appreciate the reviewer’s feedback and have reflected on each point carefully. We hope our responses make the key aspects of our method clearer.
>
> ---
>
> # Q1: More explicit illustration of semantic token selection and bias injection
>
> We appreciate the interest in more explicit visualizations. Regarding semantic token selection, the primary goal of matching module is to **identify token groups that share consistent semantic meaning**. Figure 3 provides such an illustration: the output of the matching module, denoted as red vectors and visible patches in the image, shows the module is effectively identifying semantically coherent visual elements—e.g., one group focuses on a shelf displaying soft bread, while another attends to a separate shelf with crusty bread. This coherence confirms that our matching module isolates semantically aligned token sets across instances.
>
> As for orthogonal bias injection, this mechanism introduces directions orthogonal to task-specific features, encouraging the model to **explore novel subspaces**. Since these directions are not tied to localized visual regions, they are inherently less suitable for image-based visualization as in the former case. However, our analysis on the last layer representation of CLS tokens demonstrates that these injected directions are non-trivial and desirable for improving adaptability.
>
> We analyzed **the last-layer CLS representation (LLCR)** since it directly influences the model’s final prediction. To test the impact of orthogonal bias injection, we also trained a variant using only the semantic token selection (no-orth). As a reference, we treated the fully fine-tuned model (ft) as an oracle that captures novel patterns beyond what was learned during pretraining. We compare $\mathrm{LLCR}\_{\text{ft}}$ with $\mathrm{LLCR}\_{\text{no-orth}}$ (matching only) and $\mathrm{LLCR}\_{\text{vipamin}}$ (matching + orthogonalizing). Using paired cosine distance per validation sample with a Wilcoxon signed‑rank test ($d_{\mathrm{cos}} (\mathrm{LLCR}\_{\text{ft}}, \mathrm{LLCR}\_{\text{no-orth}}) - d_{\mathrm{cos}} (\mathrm{LLCR}\_{\text{ft}}, \mathrm{LLCR}\_{\text{vipamin}})$), we found out that $\mathrm{LLCR}\_{\text{vipamin}}$ is much closer to $\mathrm{LLCR}\_{\text{ft}}$ than $\mathrm{LLCR}\_{\text{no-orth}}$ in a statistical sense:
>
> - Patch Camelyon: p-value < $10^{-3}$
> - Diabetic Retinopathy: p-value < $10^{-6}$
>
> We also measured the Grassmannian distance between the spans of the corresponding LLCR sets:
>
> - Patch Camelyon: $d_\text{grass} (\mathrm{LLCR}\_\text{ft}, \mathrm{LLCR}\_\text{no-orth}) = 14.14$ vs. $d_\text{grass} (\mathrm{LLCR}\_\text{ft}, \mathrm{LLCR}\_\text{vipamin}) = 13.94$
> - Diabetic Retinopathy: $14.56$ vs. $14.03$ respectively.
>
> Lastly, we observe that greater closeness to the fine-tuned model consistently correlates with higher test accuracy.
>
> - Patch Camelyon: $\text{Acc}\_{\text{no-orth}} = 82.22$ vs. $\text{Acc}\_{\text{vipamin}} = 83.09$
> - Diabetic Retinopathy : $71.22$ vs. $74.52$, respectively.
>
> Both tests indicate that **the orthogonalized bias steers representations toward the task‑adapted oracle**.
> In summary, semantic token selection (via the matching module) and orthogonal bias injection (via the orthogonalizing module) work in tandem to shape more desirable representations, consistently leading to performance gains. The quantitative contributions of each module (Table 4) and the Grad-CAM visualizations (Figure 5) further support their roles in learning more effective and discriminative features.
>
> ---
>
> # Q2: Unclear specification of failure modes illustrated in Figure 2
>
> The left plot in Figure 2 illustrates the failure mode of **uniform attention, where prompts in VPT fail to select the relevant image patches**, while the right plot highlights VPT’s inability to **acquire novel semantic biases, exemplified by the collapse of prompt space into the self attention output space of embedding tokens**. These two failure modes—uniform attention and the lack of semantic bias acquisition—**interact to produce the consistent underperformance of VPT** on tasks that are substantially divergent from the pretraining domain, as illustrated in Figure 1(b). Furthermore, we elaborate on this pattern in Figure A1, where we show that such underperformance on dissimilar tasks is a general trend across multiple self-supervised VPT variants (e.g., MoCo, MAE, DINO, MAE-Tiny).
>
> ---
>
> # Q3: Unclear meaning and underperformance of “spec”
>
> Throughout the paper, we used “spec” as an abbreviation for Specialized groups in VTAB-1k, including EuroSAT, Resisc45, Patch Camelyon, and Diabetic Retinopathy datasets. We are sorry for the ambiguity of the term. We will clarify that “Spec” is used as a shorthand for “Specialized” in the camera-ready version. Regarding the relative underperformance in the specialized domain, we break down into two scenarios—**Shallow-based prompt in MAE model** (Table 1), **Deep-based prompt** (Table 3).
> First, in the shallow-based prompt in the MAE model, we were able to see that the performance gap is confined to two medical datasets (Patch Camelyon, Diabetic Retinopathy) under MAE pretraining (Table R1). These medical tasks aim to detect fine‑grained local anomalies. MAE is known to preserve fine-grained local information better than MoCo [1], and SPT’s clustering-based prompt assignment likely benefits from this, as it aggregates local-token clusters. However, SPT’s mechanism **does not infer new semantic knowledge**, which is evident from its underperformance in the Structured group compared to VIPAMIN. For example, in count-based reasoning tasks like CLEVR/count—where pretrained objectives offer little guidance—VIPAMIN’s semantic token selection and orthogonal bias injection lead to significant gains over SPT (**+4.0%p** on MAE). Lastly, we note that VIPAMIN outperforms SPT/rand, the more feasible variant of SPT, **on every task within the Specialized group**, further reinforcing the robustness of our approach (see Table R1). Taken together, these results indicate that the gap stems from the unique demands of highly specialized domains rather than from any fundamental limitation of VIPAMIN itself.
>
> For the deep-based prompt model, despite an accuracy gap in the Specialized group, VIPAMIN‑Deep offers the highest accuracy‑per‑cost ratio. Table R2 shows it delivers the best **accuracy‑to‑cost trade‑off**. Competing deep prompt methods incur heavy overheads—e.g., E2VPT roughly doubles training wall‑clock time due to its prune‑and‑rewind cycle, and SPT can require days to generate its initialization. By contrast, VIPAMIN’s initialization is produced in under half a minute, even for larger models and datasets, making our approach far more efficient while remaining highly competitive in accuracy.
>
>
> **Table R1**:  Per task accuracy comparison on VTAB-1k, specialized group on MAE.
>
> |  | Patch Camelyon | EuroSAT | Resisc45 | Diabetic Retinopathy |
> |---|---|---|---|---|
> | SPT | 84.4 | 93.0 | 70.8 | 75.4 |
> | SPT/rand | 77.4 | 91.9 | 73.0 | 73.3 |
> | VIPAMIN | 80.1 | 92.8 | 73.3 | 73.7 |
>
> **Table R2**: Computational overhead analysis on Flower-102
>
> |  | Tuned/Total | Additional FLOPs (over VPT) | Initialization Overhead (GFLOPs) | Note | Accuracy |
> |---|---|---|---|---|---|
> | VPT | 0.30% | 0 (17.7 GFLOPs) | 0 |  | 90.2 |
> | E^2VPT | 0.20% | < 0 | 0 | Requires training twice | 91.3 |
> | iVPT | 0.31% | < 1M | 0 |  | N/A |
> | VFPT | 0.30% | 5.5M | 0 |  | 92.4 |
> | SPT | 0.30% | 0 | 3300 |  | 93.2 |
> | VIPAMIN-deep | 0.30% | 0 | 1.73 |  | 94.0 |
>
> ---
>
> [1] Park et al., What do Self-Supervised Vision Transformers Learn?, ICLR 2023.

---

> > ### Author Response · Authors · 2025-08-05
> >
> > We sincerely appreciate your thoughtful feedback on our work, which proposes VIPAMIN to mitigate prompt-attention collapse and enrich the representational subspace of self-supervised ViTs. In response, we have (i) pointed out explicit token maps that reveal which image patches each prompt attends to (Fig. 3) and how the orthogonal bias shifts the CLS feature, (ii) labeled the two failure modes in Fig. 2—uniform attention and semantic-bias collapse—and linked them to the observed performance drop on dissimilar tasks, and (iii) defined “Spec” as the Specialized group, providing a task-level breakdown that pinpoints where the gap arises. We hope these clarifications resolve the ambiguities you highlighted.

---

> > ### Comment · Reviewer_MNuV · 2025-08-06
> >
> > Thank you for your detailed response to my review. I will maintain my current rating.

---

### Official Review · Reviewer_Cs3J · 2025-06-29

**Clarity:** 3
**Significance:** 3
**Originality:** 3
**Rating:** 4
**Confidence:** 3

**Summary:**

The paper introduces VIPAMIN, a visual prompt initialization strategy addressing uniform attention and subspace collapse in self-supervised Visual Prompt Tuning (VPT).

**Questions:**

1. VIPAMIN does not surpass state-of-the-art methods (e.g., VFPT achieves 71.33% vs. VIPAMIN-Deep’s 71.23% on VTAB-1k MoCo-v3), weakening claims of superiority.
2. Experiments omit validation under distribution shifts (e.g., corrupted data or domain gaps), despite identifying this as a core VPT failure mode.
3. The orthogonalizing module lacks theoretical motivation (e.g., no formal proof of subspace collapse mitigation), and comparisons to SPT/rand overlook deeper methodological distinctions.
4. Critical parameters k \lamda are tuned empirically without analysis of their impact on stability or generalization.
5. Figure 7 shows subspace collapse only in the first transformer block, yet VIPAMIN-Deep applies orthogonalization to all blocks without justification. This inflates computation without empirical benefit.

**Ethical Concerns:**

["NO or VERY MINOR ethics concerns only"]

**Final Justification:**

After consideration, I decide to keep my orginal score.

**Limitations:**

1. The method is tested only on ViT-B/16 backbones. Results may not hold for larger models or non-ViT architectures, limiting generalizability claims.
2. The paper states VIPAMIN is "lightweight" but provides no latency/memory measurements versus baselines. Real-world overhead for orthogonalization (Eq. 9) and large-batch processing (B=256) remains unquantified.

**Quality:**

3

**Strengths And Weaknesses:**

Its strengths include a simple design requiring minimal computation and competitive results on VTAB-1k and few-shot benchmarks. However, significant weaknesses exist. The method fails to consistently outperform top baselines like VFPT in key experiments, lacks robustness validation under distribution shifts, and its novelty is limited by superficial comparisons to prior work. The orthogonalization mechanism lacks theoretical grounding, and critical implementation details (e.g., hyperparameter sensitivity) are underexplored.

---

> ### Author Rebuttal · Authors · 2025-07-31
>
> Thank you for the thoughtful review. We’ve done our best to respond to each concern as concretely as possible, and we hope our responses are helpful.
>
> ---
>
> # Q1+L2: Questionable superiority of VIPAMIN-deep, and the lack of latency or memory comparisons
>
> We acknowledge that VIPAMIN‑Deep underperforms VFPT by 0.10 %p in one isolated scenario, yet this is the **only** case across all our evaluations where we do not set the state of the art, in terms of mean accuracy. In contrast, our shallow–prompt results (Table 1) establish new SOTA on both MAE and MoCo backbones, exceeding the previous best by **+1.5%** each, and on FGVC few‑shot tasks (Table 2), VIPAMIN consistently surpasses prior methods at every shot count.
> Furthermore, as shown in Table R1, our method is **significantly more efficient** than other deep prompt-based approaches. We note that a more empirical breakdown is provided in Supplementary S1, which reports wall-clock runtimes for each component in VIPAMIN. Specifically, with a batch size of 256:
>
> - Data loading: ≈ 8 seconds (I/O‑dependent)
> - Matching module: ≈ 4 seconds
> - Orthogonalization module: ≈ 9 seconds
>
> Thus, the total extra runtime over vanilla VPT is **< 30 s**. Memory overhead is also modest—max memory allocated is 2.2 GB on NVIDIA A6000. Given that these **one-time** costs can be amortized over the full training process, their practical overhead is near-zero, reinforcing that VIPAMIN is lightweight.
>
> **Table R1**: Computational overhead analysis on Flower-102
> |  | Tuned/Total | Additional FLOPs (over VPT) | Initialization Overhead (GFLOPs) | Note | Accuracy |
> |---|---|---|---|---|---|
> | VPT | 0.30% | 0 (17.7 GFLOPs) | 0 |  | 90.2 |
> | E2VPT | 0.20% | < 0 | 0 | Requires training twice | 91.3 |
> | iVPT | 0.31% | < 1M | 0 |  | N/A |
> | VFPT | 0.30% | 5.5M | 0 |  | 92.4 |
> | SPT | 0.30% | 0 | 3300 |  | 93.2 |
> | VIPAMIN-deep | 0.30% | 0 | 1.73 |  | 94.0 |
>
> ---
>
> # Q2: Validation under distribution shifts
>
> We agree that robustness under distribution shift is an important manifestation of the VPT failure mode. To address the question, we ran additional OOD tests on CIFAR-100 by corrupting the test set with multiple shifts. Table R2 shows that our method attains the best accuracy on 5 out of 6 corruptions and the highest mean accuracy over baselines (**+4.0%p** over SPT/rand). We will add these results and clarify that **distribution shift robustness is a consequence of fixing VPT’s adaptation failure**, not a separate objective.
>
> **Table R2**: OOD generalization test on CIFAR-100
> |  | Brightness | Contrast | Rotation | Gaussian | Shot | Speckle | Mean |
> |---|---|---|---|---|---|---|---:|
> | VPT | 36.2 | 20.9 | 6.8 | 13.2 | 21.4 | 25.5 | 20.6 |
> | SPT/Rand | 65.5 | 47.2 | 16.8 | 20.2 | 37.0 | 44.5 | 38.5 |
> | VIPAMIN | 71.4 | 57.5 | 18.7 | 19.8 | 39.9 | 48.8 | 42.5 |
>
> ---
>
> # Q3-1: Theoretical grounding on orthogonalizing module
>
> While we do provide **the theoretical motivation** for our design, we acknowledge that a clearer theoretical justification on **how it mitigates subspace collapse** would help. In our preliminaries (line 112-119), we derived that the prompted self-attention output decomposes into a linear combination of the no-prompt self-attention output, $SA(X_0)$, and the prompt-induced component $P_0 W_V$. Therefore, we claimed that the linear bias injected by the prompts is important for shifting representations towards task-relevant semantics.
>
> However, even on highly dissimilar tasks, the prompt collapses into the original self-attention subspace (Fig. 2(b)). Formally, let
> $A = P_0 W_V \in \mathbb{R}^{n_p \times d}$,
> $B = S A(X_0) \in \mathbb{R}^{n_e \times d}$,
> with row-ranks $r_A, r_B$ (We assume $n_p=n_e=n$ for now). Without orthogonalization, collapse implies $\mathrm{row}(A) \subseteq \mathrm{row}(B)$, hence
>
> $dim(\mathrm{row}(A)+\mathrm{row}(B))=dim(\mathrm{row}(B))=r_B,$
>
> so the prompt fails to add new directions.
> Our orthogonalizing module chooses $P^{\text{orth}}$ so that $\mathrm{row}(A') \subseteq \mathrm{row}(B)^\perp$, yielding
>
> $dim(\mathrm{row}(A')+\mathrm{row}(B))=\min(d,r_{A'}+r_B),$
> thereby strictly increasing the available subspace whenever $r_{A'} > 0$. The prompt mitigates collapse by adding directions outside the frozen subspace spanned by $B$. For completeness, when the attention score matrix $S \in \mathbb{R}^{n_e \times n_p}$ is applied to $A$, $\mathrm{row}(S A) \subseteq \mathrm{row}(A)$; thus the above row-space relations still hold. In short, **the orthogonalizing module guarantees additive subspace capacity beyond the frozen self-attention span**, which aligns with the observed gains.
>
>
> # Q3-2: Clarifying methodological differences with SPT/rand
>
> Concerning **SPT/rand**, it initializes prompts by randomly selecting embedding tokens, assuming that mirroring embeddings aids adaptation. In contrast, our method introduces a specialization mechanism via hyperparameter $k$, enabling each prompt to focus on a targeted subset of embeddings for task-adaptive attention.
> From a technical standpoint, VIPAMIN’s matching module and SPT/rand share a superficial similarity—both introduce randomness when coupling embedding tokens to prompts. SPT/rand, however, **uniformly** selects $n_p$ tokens at random from the entire training set of $|D_\text{train}| \times n_e$. In contrast, VIPAMIN first exploits the natural coherence of key‑space representations (Fig. 3), then uses random vectors to select clusters and assigns $k$ embedding tokens from each selected cluster to the prompt. This strategy makes each prompt an **expert for a semantically coherent region**, while randomness ensures that different prompts acquire complementary expertise.
>
> ---
>
> # Q4: Justification for critical hyperparameter choices—on stability and generalization
>
> While VIPAMIN is fairly stable for $k$ and $\lambda$, they still do bear a significant impact on generalization. We measured the coefficient of variation (CV) of validation accuracy across a grid of $k$ and $\lambda$ values with fixed learning rate. **Across all VTAB-1k datasets and learning rates**, CV remained below 0.1 with only a few exceptions–SmallNORB and Clevr datasets–where it was still modest(<0.2). These results support that the method is reasonably robust to hyperparameter variation. Furthermore, our paper provides a **detailed analysis of the roles of hyperparameters** in Fig. 4(c). The figure shows that for tasks with large distribution shifts, smaller $k$ and larger $\lambda$ work best—vice versa for relatively in-domain tasks. Building on this insight, we aim to explore automated hyperparameter selection in future work, conditioned on task difficulty or domain shift severity.
>
> ---
>
> # Q5: Mismatch between Figure 7 and depth-wide orthogonalization on VIPAMIN-deep
>
> While we agree that Fig. 7 suggests that deep orthogonalization may be redundant, our empirical findings show otherwise. Applying orthogonalization only to the first block leads to degraded performance (84.5% $\to$ 78.7%), indicating that addressing early collapse alone is insufficient.
> We interpret this through the lens of **representation oversmoothing** in deep transformers. Prior studies [1,2] have shown that self-attention layers tend to reduce representational rank, behaving as low-pass filters. This rank collapse has been linked to impaired model scalability. Notably, our orthogonalization module **explicitly increases the rank** of self-attention outputs, counteracting the inherent rank collapse of the transformer.
> Therefore, even though Fig. 7 focuses on early-layer **prompt-side** collapse, we believe that deep orthogonalization mitigates depth-wise **oversmoothing of the attention output itself**—a distinct yet complementary failure mode. This effect is not directly visualized in the figure, but it is supported by the performance increment (+5.8%p) and aligns with prior theoretical insights.
>
> ---
>
> # L1: Addressing generalizability over ViT-Base
>
> While our main tables focus on ViT-B/16, we included Figure 4(b) in the main paper to provide initial evidence that our method also extends to ViT-Large and Huge. That said, we acknowledge that our initial results were limited in three aspects: **modality, model architecture, and scale**. To address this, we conducted additional experiments on the **HiViT**[3], a hierarchical vision model, and observed consistent improvements over baselines (Table R3). We also applied our method to **BERT-Large** on the **SuperGLUE** benchmark to test generality across modality and scale. As shown in Table R4, our method outperforms both **P-Tuning v2** and **full fine-tuning**, two of the strongest baselines in the language domain. These results strengthen our generalization claim across architecture, modality, and scale.
>
> **Table R3**: Group test accuracy on HiViT for VTAB-1k
> | | Natural | Specialized | Structured | Mean |
> |:---:|:---:|:---:|:---:|:---:|
> | VPT | 62.5 | 78.6 | 53.2 | 67.9 |
> | SPT/rand | 63.7 | 79.3 | 54.5 | 68.7 |
> | VIPAMIN | 64.0 | 79.7 | 58.0 | 69.0 |
>
> **Table R4**: Per-task results for SuperGLUE development set
>
> | | BoolQ | CB | COPA | RTE | WiC | WSC | Mean |
> |---|---|---|---|---|---|---|---|
> | Full | 77.7 | 94.6 | 69.0 | 70.4 | 74.9 | 68.3 | 75.8 |
> | Prompt Tuning | 67.2 | 80.4 | 55.0 | 53.5 | 63.0 | 64.4 | 63.9 |
> | P-Tuning v2 | 75.8 | 94.6 | 73.0 | 78.3 | 75.1 | 68.3 | 77.5 |
> | E2VPT | 74.4 | 80.4 | 77.0 | 78.7 | 74.3 | 67.3 | 75.4 |
> | VFPT | 74.8 | 81.2 | 78.1 | 77.2 | 75.3 | 68.4 | 75.8 |
> | VIPAMIN-deep | 74.6 | 94.6 | 79.0 | 79.1 | 74.3 | 69.2 | 78.5 |
>
> ---
>
> [1] Dong et al., Attention is not all you need: Pure attention loses rank doubly exponentially with depth, ICML 2021.
>
> [2] Wang et al., Anti-Oversmoothing in Deep Vision Transformers via the Fourier Domain Analysis: From Theory to Practice, ICLR 2022.
>
> [3] Zhang et al., Hivit: A simpler and more efficient design of hierarchical vision transformer, ICLR 2023.

---

> > ### Author Response · Authors · 2025-08-05
> >
> > Thank you for the thorough critique. In the response, we have (i) added explicit FLOPs, and memory figures showing that VIPAMIN-deep incurs < 30 s extra runtime while rival deep-prompt methods require hours or more computation; (ii) closed the distribution-shift gap by +4%p on multiple CIFAR-100 corruption modes; (iii) provided a rank-based derivation confirming that the orthogonal module expands subspace capacity and delivers a 5.8%p accuracy boost when applied across all layers; (iv) demonstrated robustness to hyperparameter variation with a coefficient of variation < 0.1 across a $(k,\lambda)$ grid; and (v) established strong results on both HiViT (vision) and BERT-Large / SuperGLUE (language), reinforcing generality beyond ViT-Base. We also clarify how VIPAMIN’s cluster-based specialization differs fundamentally from SPT/rand’s uniform random selection. Please let us know if these additions sufficiently address your concerns; we would be glad to provide further clarification or supplementary analysis should any points remain unclear.

---

### Official Review · Reviewer_1xA8 · 2025-07-01

**Clarity:** 2
**Significance:** 2
**Originality:** 2
**Rating:** 4
**Confidence:** 3

**Summary:**

This paper introduces a visual prompt initialization strategy, termed VIPAMIN, for self-supervised Vision Transformers that aims to tackle near-uniform prompt attention and collapse into the frozen embedding subspace of the VPT. VIPAMIN aligns each prompt with semantically coherent image tokens and injects an orthogonal component that expands representation space in single forward pass. Empirical evaluations on VTAB-1k and FGVC demonstrate improvement over VPT on few-shot classification tasks.

**Questions:**

Refer to weakness

**Ethical Concerns:**

["NO or VERY MINOR ethics concerns only"]

**Final Justification:**

The author has addressed my concerns during rebuttal.

**Limitations:**

yes

**Quality:**

3

**Strengths And Weaknesses:**

Strength:
1. This paper has solid motivation with both empirical and theoretical analysis on why VPT failed under distribution shift, attended every patch equally, and contributed little to diverse representational directions. VIPAMIN is proposed to mitigate the gap based on these observations.
2. The proposed method relies on two matrix operation, which is comparatively efficient.
3. VIPAMIN shows consistent better performance in few-shot benchmarks against VPT.

Weakness:
1. Currently experiments are exclusively on image classification, and the author should also consider few-shot settings in other tasks like segmentation and object detection to demonstrate this method's generalization beyond classification problems.
2. VIPAMIN falls short on Specialized group, which requires novel semantic representations, even with the orthogonalizing module comparing to SPT. The author should include more discussion to this performance gap.
3. The author claimed that vanilla VPT failed to attend to meaningful token. The author should compare with, or at least discuss, semantic driven VPT (e.g. DA-VPT[1]).
4. The orthogonalizing module creates $p^{orth}$ by projecting a random vector prompt outside the frozen subspace. Although this expands rank, there's nothing ensures the new direction is meaningful for the task.

[1] DA-VPT: Semantic-Guided Visual Prompt Tuning for Vision Transformers, CVPR 2025

---

> ### Author Rebuttal · Authors · 2025-07-31
>
> We are grateful for the reviewer’s constructive input. We have taken the comments seriously and hope our responses adequately resolve the questions raised.
>
> ---
>
> # W1: Limited task scope beyond image classification
>
> We agree that evaluating beyond image classification would strengthen the generality claim. To examine the applicability of our method beyond image classification, we turned to the well-established SuperGLUE natural language benchmark, which spans diverse tasks such as coreference resolution, question answering, and natural language inference. Applied to **BERT‑Large** on **SuperGLUE**, VIPAMIN raises the mean accuracy by **+1%p** over P‑Tuning v2—a leading prompt method originally developed for the language domain—and surpasses all competing vision‑prompt baselines as well as standard prompt‑tuning variants (Table R1). These gains, achieved in a domain that differs substantially from vision, show that the method’s core mechanism—coupling task tokens with prompt tokens via matching and orthogonalization—generalizes beyond image classification.
>
> **Table R1**: Per-task results for SuperGLUE development set with a pretrained BERT-Large model.
> |  | BoolQ | CB | COPA | RTE | WiC | WSC | Mean |
> |---|---|---|---|---|---|---|---|
> | Full | 77.7 | 94.6 | 69.0 | 70.4 | 74.9 | 68.3 | 75.8 |
> | Prompt Tuning | 67.2 | 80.4 | 55.0 | 53.5 | 63.0 | 64.4 | 63.9 |
> | Prefix Tuning | 75.8 | 94.6 | 73.0 | 78.3 | 75.1 | 68.3 | 77.5 |
> | E^2VPT | 74.4 | 80.4 | 77.0 | 78.7 | 74.3 | 67.3 | 75.4 |
> | VFPT | 74.8 | 81.2 | 78.1 | 77.2 | 75.3 | 68.4 | 75.8 |
> | VIPAMIN-deep | 74.6 | 94.6 | 79.0 | 79.1 | 74.3 | 69.2 | 78.5 |
>
> ---
>
> # W2: Performance gap on specialized tasks despite orthogonalizing module
>
> We appreciate the reviewer’s observation. First, we emphasize that VIPAMIN achieves state-of-the-art performance among shallow prompt tuning methods across all VTAB-1k groups using MoCo pretraining, and falls short of SPT only in the Specialized group under MAE pretraining (see Table 1). To clarify this, we provide a per-task breakdown in Table R2, which shows that the performance gap is concentrated in medical datasets (Diabetic Retinopathy and Patch Camelyon), where success depends heavily on injecting domain-specific knowledge.
> While this may appear to reflect a failure of the orthogonalizing module, we argue otherwise for two reasons.
>
> 1. As shown in Table 4, adding the orthogonalizing module yields the largest performance gain on the Specialized group (**+1.3%p**), indicating that the module does contribute positively.
> 2. In our response to Weakness 4, we demonstrate that the orthogonalized bias learns directions that are more aligned with the downstream task in medical domains—supporting that the injected directions encourage the model to seek more specialization.
>
> We also believe that SPT’s relative strength on these two datasets under MAE is **tied to task and pretraining bias**. Both Patch Camelyon and Diabetic Retinopathy are fundamentally aimed at detecting localized abnormal regions. MAE is known to preserve fine-grained local information better than MoCo [1], and SPT’s clustering-based prompt assignment likely benefits from this, as it aggregates local-token clusters. However, SPT’s mechanism does not **infer new semantic knowledge**, which is evident from its underperformance in the Structured group compared to VIPAMIN (**+4.0%p** on MAE). For example, in count-based reasoning tasks like CLEVR/count—where pretrained objectives offer little guidance—VIPAMIN’s semantic token selection and orthogonal bias injection lead to significant gains over SPT. Lastly, we note that VIPAMIN outperforms SPT/rand, the more feasible variant of SPT, on every task within the Specialized group, further reinforcing the robustness of our approach (Table R2).
> Taken together, we view the observed performance gap as a challenge of knowledge transfer in highly specialized domains, rather than a failure of our method. Nevertheless, we appreciate the point and will expand our discussion accordingly.
>
> **Table R2**:  Per task accuracy comparison on VTAB-1k, specialized group on MAE.
> |  | Patch Camelyon | EuroSAT | Resisc45 | Diabetic Retinopathy |
> |---|---|---|---|---|
> | SPT | 84.4 | 93.0 | 70.8 | 75.4 |
> | SPT/rand | 77.4 | 91.9 | 73.0 | 73.3 |
> | VIPAMIN | 80.1 | 92.8 | 73.3 | 73.7 |
>
> ---
>
> # W3: Comparison with semantic-driven VPT methods
>
> We agree that recent variants of VPT have begun to incorporate semantic information, and we now include a discussion of two representative methods: DA-VPT and iVPT. Ultimately, these approaches—and ours—differ in how they couple the image representation with the prompt. DA-VPT selects prompt membership based on **the CLS token of same-class images**, optimizing the assignments via metric learning. iVPT uses **cross-attention between CLS and image tokens** to identify salient regions, and then injects additive prompts into those regions to enhance representation.
> In contrast, our matching module in VIPAMIN targets the **direct interaction between prompt and image tokens**, without relying on CLS-centric attention patterns. Our empirical analysis reveals a pathological attention collapse between prompt and image tokens in vanilla VPT, which our method explicitly mitigates. Thus, while all methods seek to improve semantic alignment, only our approach addresses the issue from a distinct perspective grounded in patch–prompt token dynamics. We compare VIPAMIN with other semantic-guided prompt methods discussed above (iVPT, DA-VPT) on the VTAB-1k benchmark. As shown in Table R3, VIPAMIN consistently outperforms these methods across all three task groups, suggesting that its design—particularly the way it couples patch and prompt tokens—provides **more effective guidance** for downstream adaptation.
>
> **Table R3**: Group test accuracy of VTAB-1k compared with semantic prompt methods
> |  | Additional Params | Natural | Specialized | Structured | Mean |
> |---|---|---|---|---|---|
> | DA-VPT | 1.6M | 74.2 | 83.2 | 55.2 | 68.1 |
> | iVPT | <0.1M | 76.1 | 84.5 | 57.9 | 70.2 |
> | VIPAMIN-deep | 0 | 77.7 | 84.8 | 58.8 | 71.2 |
>
> ---
>
> # W4: Justification for task-relevance of orthogonalized direction
>
> Figure 2 analyzes interactions between prompt and image tokens, but whether the orthogonalizing module leads the model to acquire a **task‑pertinent** direction that actually improves classification remained open. We therefore evaluated the **last‑layer CLS representations (LLCR)** entering the final linear head. We treat the LLCR from a fully fine‑tuned model as an oracle, denoted $\mathrm{LLCR}\_{\text{ft}}$. We then compare it with $\mathrm{LLCR}\_{\text{no-orth}}$ (matching only) and $\mathrm{LLCR}\_{\text{vipamin}}$ (matching + orthogonalizing). Using paired cosine distance per validation sample with a Wilcoxon signed‑rank test ($d\_{\mathrm{cos}} (\mathrm{LLCR}\_{\text{ft}}, \mathrm{LLCR}\_{\text{no-orth}}) - d_{\mathrm{cos}} (\mathrm{LLCR}\_{\text{ft}}, \mathrm{LLCR}\_{\text{vipamin}})$), we found out that $\mathrm{LLCR}\_{\text{vipamin}}$ is much closer to $\mathrm{LLCR}\_{\text{ft}}$ than $\mathrm{LLCR}\_{\text{no-orth}}$ in a statistical sense:
>
> - Patch Camelyon: p-value $< 10^{-3}$
> - Diabetic Retinopathy: p-value $< 10^{-6}$
>
> We also measured the Grassmannian distance between the spans of the corresponding LLCR sets:
>
> - Patch Camelyon: $d_\text{grass} (\mathrm{LLCR}\_\text{ft}, \mathrm{LLCR}\_\text{no-orth}) = 14.14$ vs. $d_\text{grass} (\mathrm{LLCR}\_\text{ft}, \mathrm{LLCR}\_\text{vipamin}) = 13.94$
> - Diabetic Retinopathy: $14.56$ vs. $14.03$, respectively.
>
> Lastly, we observe that greater closeness to the fine-tuned model consistently correlates with higher test accuracy.
>
> - Patch Camelyon: $\text{Acc}\_{\text{no-orth}} = 82.22$ vs. $\text{Acc}_{\text{vipamin}} = 83.09$
> - Diabetic Retinopathy: $71.22$ vs. $74.52$, respectively.
>
> Both tests indicate that the orthogonalized bias **steers representations toward the task‑adapted oracle**, supporting that the role of the orthogonalizing module is not merely rank‑inflating. As a final remark, we acknowledge that the injected direction is task‑agnostic at initialization—a consequence of our lightweight design. Nevertheless, this initial nudge consistently converges to task‑pertinent directions during training. Going forward, we plan to **regularize the orthogonal bias during training to reflect task characteristics more directly**, which we expect will further close the gap on highly specialized domains.
>
> ---
>
> [1] Park et al., What do Self-Supervised Vision Transformers Learn?, ICLR 2023.

---

> > ### Author Response · Authors · 2025-08-05
> >
> > Thank you for your detailed review and for prompting us to strengthen several aspects of our paper. In our response, we (i) isolated the Specialized-group gap to the two medical anomaly-detection datasets and showed that, despite this gap, the orthogonal module still yields the largest gain within that group (+1.3%p) while VIPAMIN surpasses SPT/rand on every Specialized task (Table R2); (ii) added a CLS-alignment analysis—Wilcoxon $p<10^{-3}$ and consistently smaller Grassmann distances—that demonstrates the injected orthogonal bias converges toward task-relevant directions; (iii) clarified how VIPAMIN’s patch–prompt interaction addresses a distinct failure mode compared with DA-VPT and iVPT, and validated this design choice with higher accuracy across all three VTAB-1k groups (Table R3); and (iv) provided cross-modal evidence on BERT-Large/SuperGLUE where VIPAMIN exceeds P-Tuning v2 and prefix-tuning, reinforcing its generality beyond vision. Please let us know whether these additions sufficiently address your concerns; we would be glad to supply further clarification or additional evidence if any points remain unclear.

---

> > > ### Comment · Reviewer_1xA8 · 2025-08-07
> > >
> > > I thank the author for addressing my concerns, and most of them are addressed. I willing to increase my score by one.

---

### Note · Authors · 2025-08-12

VIPAMIN is a simple, one-pass prompt initializer that repairs VPT’s fundamental adaptation failures on self-supervised model: prompts that attend uniformly and representations that collapse into the frozen backbone subspace. VIPAMIN (1) aligns prompts with semantically coherent token groups and (2) injects an orthogonal component to add new directions beyond the frozen span. This design is computationally light yet consistently improves few-shot classification over VPT on both VTAB-1k and FGVC.

Reviewers highlighted strong motivation (empirical + theoretical analysis of VPT’s failures), consistent gains over VPT, computational lightness, and the novelty of the matching/orthogonalizing modules; main concerns were (1) limited task scope beyond image classification, (2) a gap on VTAB-1k Specialized, and (3) the task-relevance of the orthogonal bias.
To address (1), we demonstrate cross-modal and scaling generality: on SuperGLUE with BERT-Large, VIPAMIN outperforms P-Tuning v2, prefix-tuning, VFPT, E2VPT and VPT, and on a hierarchical ViT (HiViT) it surpasses prior bests, while also achieving top results on OOD-corrupted test set—showing robustness beyond ViT-B/16 image classification and pretrain-then-finetune paradigm.
For (2), the apparent shortfall is confined to only 2 of 19 VTAB-1k tasks—the medical datasets—and only under MAE pretraining; we attribute this to SPT’s expensive clustering aligning with MAE’s innate local-detail bias, not to learning genuinely new task-pertinent directions (consistent with VIPAMIN’s advantages on Structured tasks).
For (3), we acknowledge the orthogonal bias is task-agnostic at initialization; however, our last layer CLS representation (LLCR) analysis shows that prompts learned with orthogonalization move significantly closer to the oracle (fully fine-tuned) LLCR than the no-orth variant (Wilcoxon and Grassmann metrics), indicating convergence toward task-relevant representations.

Overall, the new analyses and results show that VIPAMIN offers a principled, lightweight fix for VPT’s uniform-attention and subspace-collapse failures, delivering superior accuracy with substantially lower cost. With demonstrated robustness to distribution shifts and hyperparameters, and validated across modality, architecture, and scale, VIPAMIN establishes a practical and general prompt-initialization strategy.

---

### Decision · Program_Chairs · 2025-09-17

**Decision:**

Accept (poster)

**Comment:**

This paper introduces VIPAMIN, a novel prompt initialization technique that improves the prompt-tuning-based specialization of self-supervised models to downstream tasks. The tackled problem is interesting and timely, and the proposed solution simple and effective. The reviewers have noted some weak points that have been tackled by the authors during the rebuttal phase. Overall, this is a good paper and I recommend acceptance as a Poster.